# Expandable and reversible copy number amplification drives rapid adaptation to antifungal drugs

Robert T Todd, Anna Selmecki*

Department of Microbiology and Immunology, University of Minnesota Medical School, Minneapolis, Minnesota, United States

**Abstract** Previously, we identified long repeat sequences that are frequently associated with genome rearrangements, including copy number variation (CNV), in many diverse isolates of the human fungal pathogen *Candida albicans* (Todd et al., 2019). Here, we describe the rapid acquisition of novel, high copy number CNVs during adaptation to azole antifungal drugs. Single-cell karyotype analysis indicates that these CNVs appear to arise via a dicentric chromosome intermediate and breakage-fusion-bridge cycles that are repaired using multiple distinct long inverted repeat sequences. Subsequent removal of the antifungal drug can lead to a dramatic loss of the CNV and reversion to the progenitor genotype and drug susceptibility phenotype. These findings support a novel mechanism for the rapid acquisition of antifungal drug resistance and provide genomic evidence for the heterogeneity frequently observed in clinical settings.

## Introduction

The evolution of antifungal drug resistance is an urgent threat to human health worldwide, particularly for hospitalized and immune-compromised individuals (*Perea and Patterson, 2002*; *Pfaller, 2012*; *Vandeputte et al., 2012*). Only three classes of antifungal drugs are currently available and resistance to all three classes occurred for the first time in the emerging fungal pathogen *Candida auris* (*Chen and Sorrell, 2007*; *Ghannoum and Rice, 1999*; *Lockhart et al., 2017*). Importantly, the mechanisms and dynamics of acquired antifungal drug resistance, in vitro or in a patient undergoing antifungal drug therapy, are not fully understood.

The most common human fungal pathogen, *Candida albicans*, causes nearly 500,000 life-threatening infections each year (*Brown and Netea, 2012*). Disseminated bloodstream infections of *C. albicans* have a high mortality rate (15–50%) despite available antifungal therapies (*Pfaller et al., 2010*; *Pfaller et al., 2019*). The failure of antifungal drug therapy is likely multifactorial and is compounded by the fungistatic, not fungicidal, mechanisms of most antifungal drugs (*Bicanic et al., 2009*; *Roemer and Krysan, 2014*). Additionally, antifungal drug tolerance, the fraction of growth above an individual isolate's minimum inhibitory concentration (MIC), can cause an inability to effectively clear these fungal infections (*Berman and Krysan, 2020*). Mechanisms that cause antifungal drug tolerance are not fully understood, but likely include the induction of cell growth and division, core stress response regulators, and cell wall and cell membrane biosynthesis pathways (*Berman and Krysan, 2020*; *Mayer et al., 2013*; *Onyewu et al., 2004*; *Rosenberg et al., 2018*; *Sanglard et al., 2003*).

*C. albicans* and other fungal pathogens exhibit significant karyotype and genome plasticity (*Bravo Ruiz et al., 2019*; *Chibana et al., 2000*; *Croll and McDonald, 2012*; *Gerstein et al., 2015*; *Magee and Magee, 2000*; *Selmecki et al., 2010*; *Shin et al., 2007*; *Sionov et al., 2010*; *Suzuki et al., 1982*; *Zolan, 1995*). The genome plasticity observed in *C. albicans* isolates is somatic (asexual), based upon the absence of evidence for a meiotic cell cycle (*Alby et al., 2009*;

*For correspondence:
selmecki@umn.edu

Competing interests: The authors declare that no competing interests exist.

*Forche et al., 2008*; *Hull and Johnson, 1999*; *Magee and Magee, 2000*; *Tzung et al., 2001*), and includes whole genome duplication/reduction, aneuploidy, segmental aneuploidy, and loss of heterozygosity (LOH) (*Abbey et al., 2014*; *Ene et al., 2018*; *Forche et al., 2008*; *Forche et al., 2019*; *Ford et al., 2015*; *Gerstein et al., 2017*; *Hickman et al., 2013*; *Hirakawa et al., 2015*; *Ropars et al., 2018*; *Rustchenko-Bulgac, 1991*; *Selmecki et al., 2006*; *Todd et al., 2017*). From an evolutionary prospective, the genome plasticity of *C. albicans* (and other fungal pathogens) may dramatically alter the frequency with which beneficial mutations are acquired within a population, resulting in both drug resistance and drug tolerance phenotypes.

Genome plasticity due to amplification or deletion of a chromosome segment, defined herein as copy number variation (CNV), is found across all domains of life (*Anderson and Roth, 1977*; *Beroukhim et al., 2010*; *Chow et al., 2012*; *Dulmage et al., 2018*; *Elde et al., 2012*; *Riehle et al., 2001*; *San Millan et al., 2017*; *Zarrei et al., 2015*; *Żmieńko et al., 2014*). CNVs are highly prevalent in human cancers, resulting in tumorigenesis, metastasis, and increased rates of mortality (*Beroukhim et al., 2010*; *Heitzer et al., 2016*; *Hieronymus et al., 2018*; *Shlien and Malkin, 2009*; *Zack et al., 2013*). In *Saccharomyces cerevisiae*, CNVs of membrane transporters (e.g. *HXT6/7, CUP1, GAP1,* and *SUL1*) can provide a strong fitness benefit in nutrient limiting or high-copper environments (*Adamo et al., 2012*; *Brown et al., 1998*; *Gresham et al., 2008*; *Gresham et al., 2010*; *Hull et al., 2017*; *Lauer et al., 2018*; *Lin and Li, 2011*; *Payen et al., 2014*; *Selmecki et al., 2015*). Many CNVs occur due non-allelic homologous recombination (NAHR) between repeat sequences (*Chow et al., 2012*; *Deng et al., 2015*; *Finn and Li, 2013*; *Haber and Debatisse, 2006*; *Hastings et al., 2009*; *Lobachev et al., 2002*; *Mizuno et al., 2013*; *Narayanan et al., 2006*; *Putnam et al., 2014*; *Ramocki et al., 2009*; *Zhao et al., 2014*). Some of these CNVs are amplified via a mechanism that relies on short repetitive sequences and aberrant base pairing during replication fork stalling (*Brewer et al., 2015*; *Brewer et al., 2011*) or DNA re-replication (*Finn and Li, 2013*; *Green et al., 2010*). In both *S. cerevisiae* and human cancers, extrachromosomal circular DNA (eccDNA) can also yield high copy CNVs (*Gresham et al., 2010*; *Hull et al., 2019*; *Libuda and Winston, 2006*; *Møller et al., 2018*; *Møller et al., 2015*; *Paulsen et al., 2018*; *Singh and Wu, 2019*). Experimental evolution supports that CNVs can occur and spread rapidly within a population under selection, and competition between distinct CNV lineages (clonal interference) is frequently observed (*Lauer et al., 2018*; *Payen et al., 2014*). Additionally, CNVs can increase the rate in which de novo mutations are acquired relative to the rest of the genome, and further alter the mutational and adaptive landscape of viral, bacterial, and eukaryotic organisms (*Bayer et al., 2018*; *Cone et al., 2017*; *Elde et al., 2012*; *Otto, 2007*; *Pavelka et al., 2010*; *Sun et al., 2009*; *Yona et al., 2012*; *Zhou et al., 2011*). However, in the absence of selective pressure, CNVs are generally thought to confer a fitness defect to the cell and are removed from the population (*Adler et al., 2014*; *Tang and Amon, 2013*). Therefore, the mechanism and dynamics of CNV gain and loss are critical to our understanding of adaptive evolution and pathogenesis.

Antifungal drug stress selects for aneuploidy and CNV formation in diverse human fungal pathogens, including *C. albicans, C. auris,* and *Cryptococcus neoformans* (*Gerstein et al., 2015*; *Hwang et al., 2017*; *Muñoz et al., 2018*; *Selmecki et al., 2006*; *Selmecki et al., 2009*; *Sionov et al., 2010*). In *C. albicans,* a recurrent CNV that amplifies the entire left arm of Chr5 in an isochromosome (i(5L)) is sufficient to cause resistance to azole antifungal drugs (*Selmecki et al., 2006*; *Selmecki et al., 2008*). This resistance is due to copy number amplification of two genes on Chr5L: *TAC1,* a transcriptional activator of the multidrug transporters (*Cdr1* and *Cdr2*), and *ERG11,* the target of the azole antifungal drugs. i(5L) is frequently identified in clinical isolates and is the only CNV known to cause azole resistance in different genetic backgrounds (*Ford et al., 2015*; *Selmecki et al., 2006*; *Selmecki et al., 2008*; *Todd et al., 2019*). In addition to copy number amplification, acquisition of non-synonymous, gain-of-function mutations in *TAC1* and *ERG11* can cause resistance due to constitutive activation of drug efflux pumps and a decreased affinity to the azole drug (*Coste et al., 2004*; *Morio et al., 2010*). LOH of these gain-of-function alleles can increase the level of resistance even more dramatically (*Coste et al., 2007*; *Coste et al., 2006*; *Ford et al., 2015*; *Sanglard et al., 2003*; *Selmecki et al., 2008*; *White, 1997*).

Long repeat sequences (65 bp – ~6.5 kb) represent a significant source of genome plasticity in *C. albicans* isolates obtained both in the presence and absence of antifungal drugs (*Todd et al., 2019*). All CNV breakpoints and many LOH breakpoints occur at long repeat sequences (*Todd et al., 2019*). Furthermore, CNV and LOH breakpoints frequently co-occur within the same long repeat sequences. What is not clear is whether a conserved mechanism might link CNV and LOH events during antifungal drug selection, given that both are important mechanisms of acquired antifungal drug resistance.

Through the study of *C. albicans* isolates subjected to azole antifungal drugs, we have identified a novel mechanism driving the rapid and recurrent formation of high copy CNVs. These CNVs amplify large genomic regions to more than 12 copies per genome and decrease sensitivity to multiple antifungal drugs. CNV formation appears to occur via a dicentric chromosome intermediate and successive breakage-fusion-bridge cycles that are repaired using two distinct long repeat sequences. This mechanism promotes rapid and amplifiable CNV formation during antifungal drug selection. Once the selection is relaxed, cells with the CNV can rapidly return to the progenitor copy number, leaving little evidence that the CNV ever occurred. The transient nature of these CNVs causes phenotypic and population-level heterogeneity that is often observed with clinical isolates in the presence of antifungal drug, including: heteroresistance, trailing growth, and tolerance (*Berman and Krysan, 2020*; *Colombo et al., 2014*; *Rueda et al., 2017*). Ultimately, these CNVs represent a previously uncharacterized, complex mechanism of gene amplification and chromosome plasticity that is exploited during adaptation to antifungal drug stress.

## Results

### Extensive copy number amplifications occur during adaptation to antifungal stress

To identify mechanisms driving CNV formation during adaptation to antifungal drug stress, we conducted 48 parallel in vitro evolution experiments with four drug-susceptible *C. albicans* clinical isolates, each representing a distinct genetic background (SC5314, P75016, P75063, and P78042, *Supplementary file 1*; *Hirakawa et al., 2015*). After 100 generations in physiological concentrations of the most commonly prescribed antifungal drug, fluconazole (FLC, 1 µg/ml) (*Felton et al., 2014*), the minimum inhibitory concentration (MIC) was determined (See Materials and methods). A sub-set of the FLC-evolved isolates (14/48) that acquired at least a twofold increase in $MIC_{50}$ relative to their progenitor were selected for whole genome sequencing (WGS). WGS analysis revealed that all (14/14) FLC-evolved isolates acquired one or more whole chromosome and/or segmental chromosome aneuploidies (*Figure 1A*). Half of the isolates (7/14) acquired novel CNVs (referred to herein as 'complex CNVs') that shared several key features: they had high copy numbers (up to 13 copies per genome), occurred entirely within a single chromosome arm (e.g. within Chr1R in AMS4107 and Chr4L in AMS4702), and were not associated with any centromere (*CEN*) sequence or the *C. albicans* repetitive element known as the Major Repeat Sequence (MRS, found eight times within the *C. albicans* genome) (*Chibana et al., 1994*; *Chindamporn et al., 1998*; *Lephart and Magee, 2006*). These complex CNVs amplified chromosome segments that ranged in length from ~164 kb to ~1.02 MB and contained 57–462 ORFs. The remaining isolates acquired either whole chromosome aneuploidies and/or segmental aneuploidies (e.g. i(5L) in AMS4106 and Chr7R in AMS4444) that had typical copy numbers (3–4 copies) and copy number breakpoints (e.g. *CEN5* in AMS4106 and *MRS7b* in AMS4444) that have been observed previously in drug-evolved isolates (*Selmecki et al., 2006*; *Todd et al., 2019*), and were not analyzed further.

### Complex CNVs are flanked by distinct long inverted repeat sequences

To identify the mechanism driving the formation of the novel complex CNVs, we determined the copy number breakpoints associated with each of the complex CNVs using a combination of read depth and allele ratio analyses. All copy number breakpoints occurred within 2 kb of one of the 1974 long repeat sequences identified previously by *Todd et al., 2019* (*Supplementary file 2*). However, unlike previous observations, these complex CNVs were comprised of a two-sided, stair-step amplification pattern that involved at least two distinct long inverted repeat sequences (*Supplementary file 2*). Accordingly, each long repeat sequence was associated with two distinct

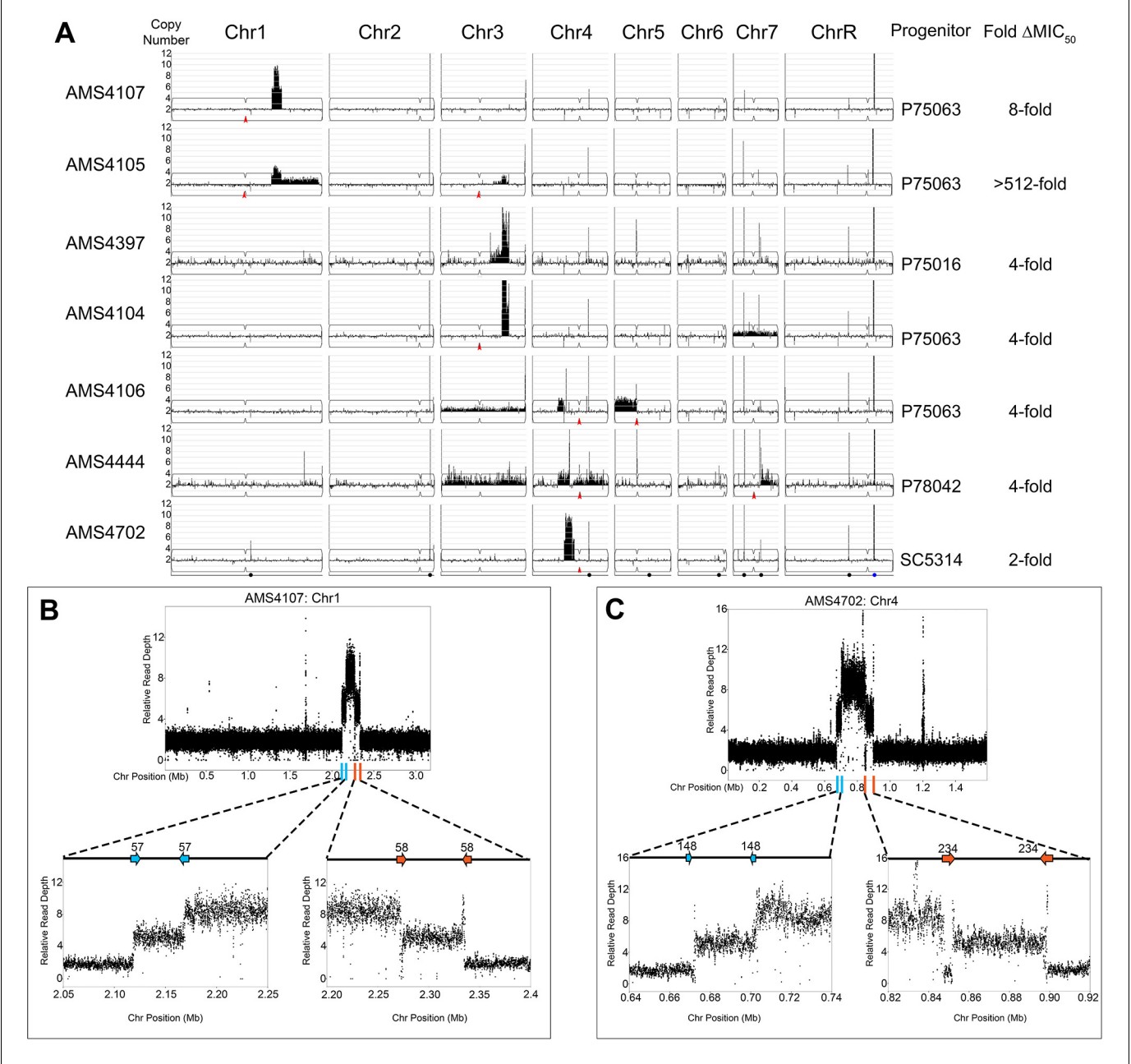

**Figure 1.** Complex CNVs are comprised of stair-step amplifications flanked by distinct long inverted repeat sequences. (A) Whole genome sequence data of FLC-evolved isolates plotted as log2 ratio and converted to chromosome copy number (y-axis, 1–12 copies) and chromosome position (x-axis, Chr1-R) using the Yeast Mapping Analysis Pipeline (YMAP). Complex CNVs amplify >12 copies of a chromosomal region ranging from 164 kb to 1.02 Mb in length. Centromeres indicated with a red arrowhead. Chromosomal positions of all MRS sequences (black dots) and the rDNA array (blue dot) are indicated below AMS4702. The progenitor of each FLC-evolved isolate and the fold increase in FLC $MIC_{50}$ at 48 hr between the progenitor and the FLC-evolved isolate is indicated. The symmetric stair-step CNV breakpoints for two isolates: (B) Chr1R of AMS4107 and (C) Chr4L of AMS4702. The relative genome sequence read depth is plotted according to chromosome position using R. The left and right side of each CNV (indicated with blue and orange lines along the x-axis of each chromosome) are expanded for higher resolution (left and right lower panels). In both isolates (AMS4107 and AMS4702), the CNV is flanked by two, distinct long inverted repeat sequences (blue and orange arrows) that do not share homology. The highest copy number amplification occurs between the two, distinct inverted repeats. All copy number breakpoints and long inverted repeat sequence details are found in *Supplementary file 2*. All genes found within the amplified regions are found in *Supplementary file 4*. Repeat numbers refer to *Supplementary file 2* from *Todd et al., 2019*.

The online version of this article includes the following figure supplement(s) for figure 1:

**Figure supplement 1.** Recurrent CNV breakpoints located on Chr3R amplify *MRR1*.

*Figure 1 continued on next page*

*Figure 1 continued*

**Figure supplement 2.** Complex CNVs increase chromosome size.
**Figure supplement 3.** Complex CNVs are predominantly found in homozygous sequence.

copy number changes, generating regions with different degrees of amplification, with the highest copy number always flanked by lower copy numbers (*Figure 1B and C*, *Supplementary file 6*). For example, a complex CNV on Chr1R (in AMS4107) comprised a ~218 kb region amplified to nine copies, flanked on the left and right by a region of variable length (the intra-repeat spacer length) that was amplified to six copies, that, in turn, was surrounded on both sides by a region of 2N copy number (the basal chromosome copy number). Surprisingly, the same symmetric stair-step copy number pattern (2-6-9-6-2 copies) was observed on two different chromosomes (Chr1 and Chr4) in two different genetic backgrounds (AMS4107 and AMS4702), indicating that the mechanism is neither chromosome-specific nor strain-specific (*Figure 1B and C*). Complex CNVs with asymmetric stair-step copy numbers also occurred due to an additional breakpoint in a third distinct long inverted repeat sequence (e.g. 2-4-6-5-3-3-2 copies in AMS4105 and 2-3-3-6-9-6-2 copies in AMS4397), that also increased the length of the CNV (see Materials and methods for determination of copy number intervals, *Supplementary file 6*). Of the eight complex CNVs, the highest copy number identified (2-7-13-7-2 copies) occurred in AMS4104 on Chr3R. These findings indicate that the mechanism driving the formation of these complex CNVs uses long repeat sequences, that the amplified number of copies is variable, and that both odd and even numbers of amplified copies can be detected.

Amplification of the same chromosomal region also occurred in isolates with different genetic backgrounds and from independent evolution experiments (*Figure 1A*). These recurrent CNVs highlight genes that are likely under selection during adaptation to FLC. For example, recurrent CNV breakpoints on Chr1R occurred at Repeats 57 and 58 in AMS4107 and AMS4105, and recurrent breakpoints on Chr3R occurred at Repeats 134 and 136 in AMS4105, AMS4397, and AMS4104 (*Figure 1—figure supplement 1*, *Supplementary file 2*). Interestingly, while the CNV breakpoints on Chr4L were not recurrent and instead occurred at different long repeat sequences, all three of the CNVs amplified a common ~118 kb region in AMS4106, AMS4444, and AMS4702 (*Supplementary file 2*). Therefore, during FLC selection, complex CNV formation results in copy number amplification of recurrent chromosomal regions.

All the long inverted repeat sequences associated with the complex CNVs were contained within a single chromosome arm (intra-chromosome arm). The occurrence of intra-chromosome arm inverted repeat sequences is relatively low (77/233) within the *C. albicans* genome (excluding MRSs and ORFs which contain complex embedded tandem repeats, see *Todd et al., 2019*). The repeat sequences found at these complex CNV breakpoints were among the longest (median 1513 bp) repeats, and had among the highest median shared sequence identity (96.1%) of all long repeat sequences found throughout the genome (median 516 bp, 95.1% median shared sequence identity), which is similar to repeats associated with breakpoints resulting in CNV, LOH and large chromosomal inversions in *C. albicans* isolates obtained in the presence and absence of antifungal drugs (*Todd et al., 2019*).

To ask if these complex CNVs were intra-chromosomal or extra-chromosomal amplifications (e.g. eccDNA that appear in in budding yeast and cancer cells; *Hull et al., 2019*; *Møller et al., 2018*; *Møller et al., 2015*; *Paulsen et al., 2018*; *Wu et al., 2019*), we used CHEF karyotype analyses. Separation of Chrs 4–7 identified a dramatic increase in Chr4 size in two isolates (AMS4702 and AMS4444) with Chr4L CNVs relative to their progenitors and indicates that these CNVs are intra-chromosomal, rather than extra-chromosomal, amplifications (*Figure 1—figure supplement 2*). Because detection of CNVs on Chr1 and Chr3 was obviated by the large size of these chromosomes, restriction digest was used to characterize these intra-chromosomal amplification events (see below, e.g. Figure 3). Therefore, while we cannot completely rule out the possibility of eccDNA amplifications in some of the isolates, the increased chromosome sizes (*Figure 1—figure supplement 2*) support the idea that the complex CNVs are intra-chromosomal amplification events.

Surprisingly, these CNVs tended to occur within chromosomal regions that were homozygous in the progenitor isolates, making it impossible to determine which haplotype was amplified in the CNVs (*Figure 1—figure supplement 3*). Nonetheless, for the CNVs from heterozygous progenitor

sequences, the amplifications derived from only one haplotype and the allele ratio scaled with copy number (e.g. the percent majority allele was ~50–62–71–64–50 for a 2-3-4-3-2 copy CNV in AMS4106). This supports that the complex CNVs arose via a stair-step mechanism that amplified just one haplotype.

## Complex CNVs increase multidrug fitness and tolerance

While bacteria usually exhibit a fitness tradeoff between drug resistance and fitness (*Bagel et al., 1999*; *Basra et al., 2018*; *Melnyk et al., 2015*), the situation is far less clear in fungal pathogens. To explore this issue comprehensively, we performed growth curve and MIC analyses across the genetically diverse isolates that had been evolved in vitro. In rich medium, the growth rate and maximum $OD_{600}$ was similar between all FLC-evolved isolates and their progenitors (*Figure 2A–D*, left panels), with several notable increases in lag phase length (growth curve summary statistics provided in *Supplementary file 3*). In the presence of FLC (1 µg/ml), the growth rate and maximum $OD_{600}$ were increased for all FLC-evolved isolates relative to their progenitors, and one isolate (AMS4444) grew better in FLC than in rich medium (*Figure 2A–D*, right panels, *Supplementary file 3*). While each progenitor and FLC-evolved isolate had unique growth trajectories, these observations support, in general, that the complex CNVs provided increased fitness in the presence of drug without a major cost to fitness in the absence of drug.

The sub-inhibitory concentrations of FLC used in our evolution experiment and the fungistatic nature of azole antifungal drugs can select for mutants that exhibit drug tolerance: the ability to grow at drug concentrations above the $MIC_{50}$ after 24 hr (*Berman and Krysan, 2020*; *Delarze and Sanglard, 2015*; *Rosenberg et al., 2018*; *Sanglard et al., 2003*). We measured tolerance to FLC and four other azole drugs (miconazole, itraconazole, ketoconazole, and posaconazole) as supra-MIC growth (SMG), which is calculated from the average growth ($OD_{600}$) at 48 hr for all wells above the $MIC_{50}$ at 24 hr (*Berman and Krysan, 2020*; *Rosenberg et al., 2018*). Four of the seven isolates with complex CNVs had higher SMG levels than their progenitors in FLC, and all seven had higher SMG levels in miconazole (*Figure 2E and F*, *Figure 2—figure supplement 1*). AMS4444 and AMS4105 had the highest SMG level in all azoles tested (0.53–0.73) and exhibited the highest growth rates and maximum $OD_{600}$ in 1 µg/ml FLC (*Figure 2C–D*). Therefore, all isolates that acquired a complex CNV had increased fitness (growth rate and $MIC_{50}$) and increased tolerance (SMG) to one or more azole antifungal drugs, and several isolates had increased tolerance to all azoles tested.

## Genes involved in drug resistance and tolerance are amplified in recurrent CNVs

To identify genes located within the CNVs that could be driving the increased fitness in azole drugs, we first characterized genes known to cause drug resistance. We started by looking at genes with the potential to encode efflux pumps. We found that *MRR1*, the transcriptional regulator of the multidrug efflux pump Mdr1, was amplified by up to 13 copies in the three isolates with a Chr3R CNV (*Figure 1—figure supplement 1*). Gain-of-function mutations in *MRR1* can cause constitutive upregulation and multidrug resistance (*Dunkel et al., 2008*; *Morschhäuser et al., 2007*; *Schubert et al., 2008*); copy number amplification alone was not previously shown to cause resistance. Additionally, multidrug transporters *CDR1* and *CDR2* map to Chr3R near *MRR1,* and these genes were amplified within the same CNV as *MRR1* in two isolates (AMS4105 and AMS4397). *MRR1* was always amplified at the highest copy number of the CNVs, while *CDR1* and *CDR2* were amplified to lower copy numbers. Finally, two other genes within the complex CNVs, *QDR2* (Chr3R) and *orf19.4889* (Chr1L), both encode predicted major facilitator superfamily (MFS) membrane transporters that may have a role in antifungal drug efflux.

Next, we searched for genes that are required for membrane and cell wall integrity, calcium and iron availability, and core stress response pathways which are likely to be important for antifungal tolerance (*Berman and Krysan, 2020*; *Garnaud et al., 2018*; *O'Meara et al., 2017*; *Onyewu et al., 2004*; *Rosenberg et al., 2018*; *Sanglard et al., 2003*; *Taff et al., 2013*). The calcineurin-regulated transcription factor *CRZ1*, involved in maintenance of membrane integrity and antifungal drug tolerance (*Onyewu et al., 2004*; *Sanglard et al., 2003*), also maps to Chr3R and was amplified along with *MRR1, CDR1*, and *CDR2* in the same two isolates mentioned above, both of which had

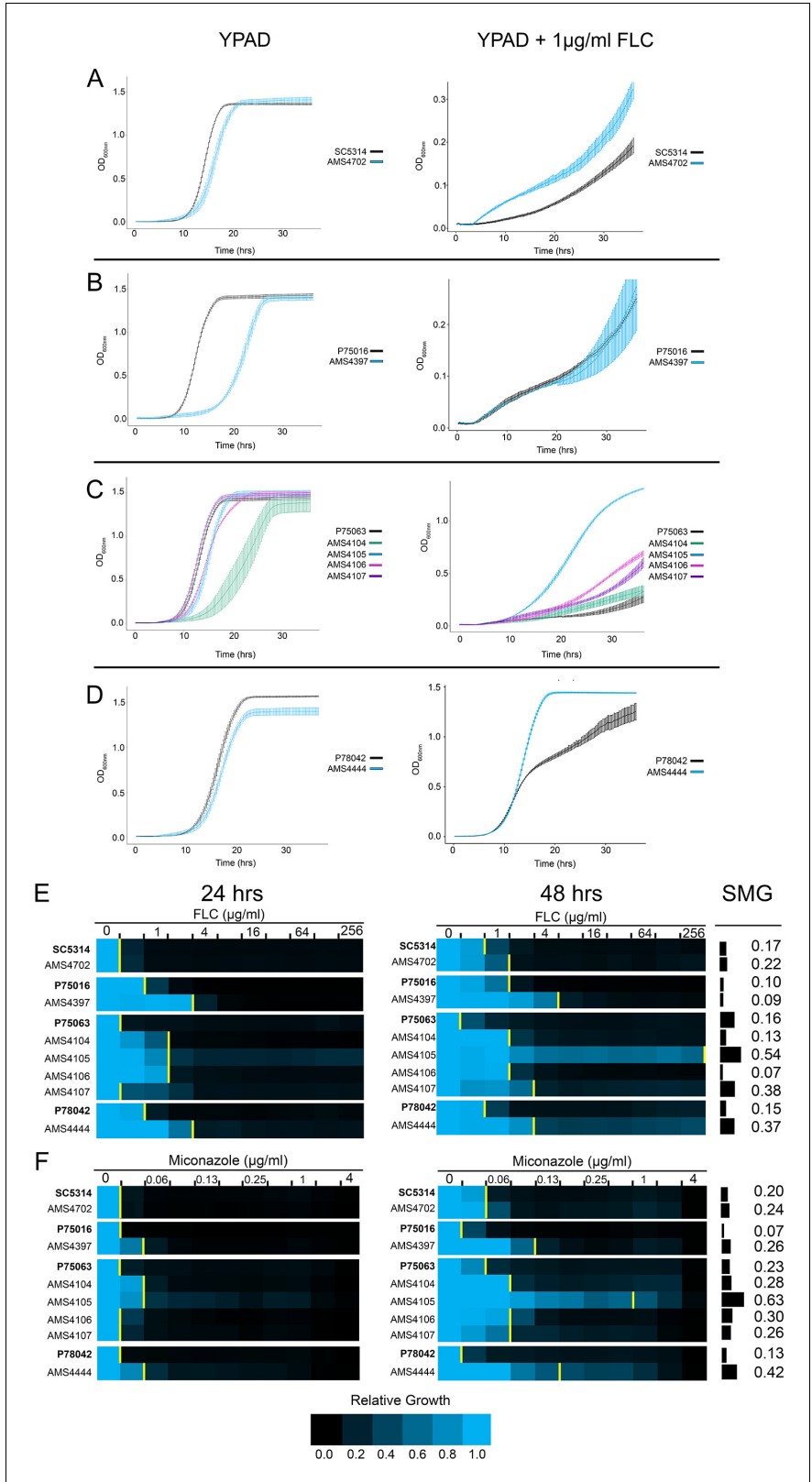

**Figure 2.** Complex CNVs increase multidrug fitness and tolerance. (**A–D**) 36 hr growth curve analysis in the absence (YPAD, left) and presence of FLC (YPAD + 1 µg/ml FLC) for each progenitor (black) and FLC-evolved isolate with a complex CNV. Average slope and standard error of the mean for three biological replicates is indicated. (**E–F**) Heat map of isolate growth ($OD_{600}$ at 24 and 48 hr) in two-fold increasing concentrations of the

*Figure 2 continued on next page*

*Figure 2 continued*

azole antifungal drugs (**E**) fluconazole (FLC) and (**F**) miconazole. The drug concentration at which 50% of growth is inhibited ($MIC_{50}$) is denoted with a yellow line on the heat map. Each heat map represents the average of three independent $MIC_{50}$ assays. Supra-MIC growth (SMG), a measurement of tolerance, was calculated as the average growth at 48 hr above the $MIC_{50}$ at 24 hr divided by the growth at 48 hr in no drug (see Materials and methods, *Figure 2—source data 1*).

The online version of this article includes the following source data and figure supplement(s) for figure 2:

**Source data 1.** Minimum inhibitory concentration raw data.
**Figure supplement 1.** Complex CNVs increase tolerance and reduce susceptibility to multiple azole drugs.

---

increased tolerance and resistance to multiple azole drugs (AMS4105 and AMS4397, *Supplementary file 4*). Other genes that encode other stress response proteins (*HSP70, CGR1, ERO1, TPK1, ASR1,* and *PBS2)* and proteins involved in membrane and cell wall integrity (*CDR3, NCP1, ECM21, MNN23, RHB1* and *KRE6*) were amplified in the CNVs (*Supplementary file 4*).

No correlation between the copy number of specific genes and the $MIC_{50}$ or SMG observed was evident. However, since these CNVs occurred in different genetic backgrounds, that differ by ~100,000 unique SNVs, the amplification of certain alleles is likely to affect fitness differently in each isolate. For isolates from the same genetic background, the copy number of each CNV and presence of additional chromosome aneuploidies may further impact fitness. The most striking increase in FLC tolerance (SMG 0.54) was observed for an isolate (AMS4105) that acquired two different CNVs: amplification of Chr3R (containing *MRR1, CDR1, CDR2, QDR2, CRZ1,* etc.) and Chr1L (containing *ORF19.4889,* etc). A different isolate (AMS4104) from the same genetic background had the same $MIC_{50}$ at 24 hr (as AMS4105), but no increase in tolerance (SMG = 0.13) relative to the progenitor (SMG = 0.16). AMS4104 acquired only a Chr3R CNV containing *MRR1*, but no amplification of *CDR1, CDR2, QDR2, CRZ1,* or the Chr1L CNV. Therefore, amplification of the *MRR1*-containing CNV correlates with an increase in the $MIC_{50}$, while amplification of additional genes in AMS4105 on Chr3R and Chr1L (including *CDR1, CDR2,* and *CRZ1*) appear to have a greater impact on tolerance.

To ask if any gene functions were enriched within the complex CNVs, we performed gene ontology (GO) analysis (*Supplementary file 4*). The cellular component 'nuclear microtubule' was the only GO term significantly enriched for all genes located within the complex CNVs (p<0.008, Bonferroni correction for multiple comparisons). The 'nuclear microtubule' term included genes that encode microtubule-binding proteins involved in spindle elongation, organization, and stabilization (*ASE1, KIP3,* and *BIM1*), chromatin remodeling (*ISW2*), and ribosome biogenesis (*NEP1*) (*Côte et al., 2009*; *Enjalbert et al., 2006*; *Eschrich et al., 2002*; *McCoy et al., 2015*; *Nobile et al., 2003*; *Nobile et al., 2012*; *Singh et al., 2011*; *Tuch et al., 2008*; *Supplementary file 4*). In summary, the complex CNVs amplified genes known to have a direct role in drug resistance (*MRR1, CDR1,* and *CDR2*) and drug tolerance (*CZR1*), as well as other roles that may be under selection during adaptation to antifungal drug stress, including maintenance of mitotic spindle function.

## Expandable CNVs generated through a step-wise amplification of a dicentric chromosome

The identification of recurrent CNV breakpoints in different genetic backgrounds raised the possibility that a common mechanism was driving the formation of these complex CNVs. To further address the mechanism of formation and to understand the impact of copy number on fitness, we used a set of isogenic isolates obtained from an agar plate instead of liquid cultures. These four single colony isolates (AMS3051, AMS3052, AMS3053, and AMS3054) were obtained from the same progenitor (AMS3050) after 120 hr growth on a single miconazole (20 µg/ml) agar plate (*Mount et al., 2018*). Prior whole genome sequencing analysis of these colonies identified a shared CNV of Chr3L (*Mount et al., 2018*). To test the hypothesis that these single colonies represented different outcomes from the same recombination event, due to the short exposure to miconazole and similarity in karyotypes, we first performed read depth analysis to characterize the CNV breakpoints. All four colonies were monosomic from the Chr3L left telomere to a shared CNV breakpoint at a long inverted repeat (Repeat 124 [blue lines], *Figure 3A*). Strikingly, the major difference between the four colonies was the maximum number of copies (3–14 copies) of the adjacent ~146 kb region on

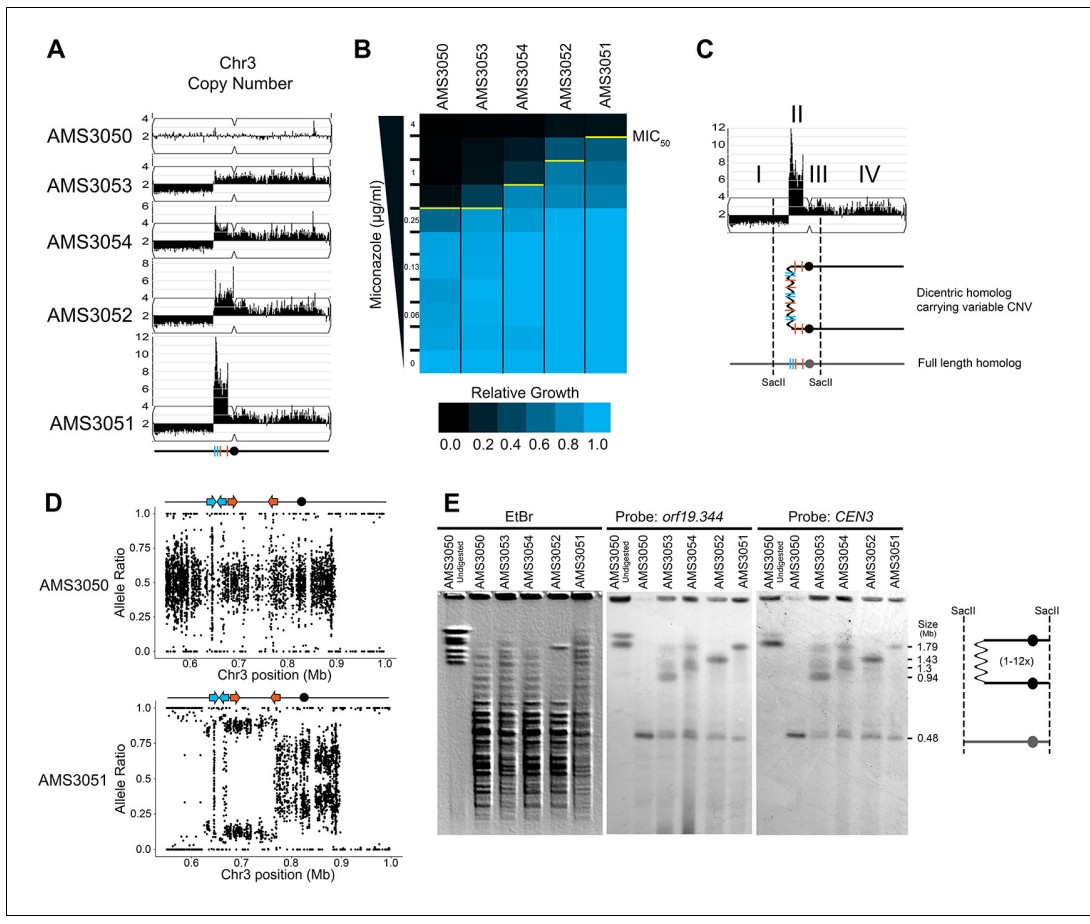

**Figure 3.** Complex CNVs are rapidly expandable in the presence of antifungal drug. (A) Whole genome sequence data of progenitor isolate (AMS3050) and miconazole-evolved single colonies (AMS3053, AMS3054, AMS3052, and AMS3051) plotted as in *Figure 1A*. All four colonies are monosomic from the Chr3L telomere to a CNV breakpoint at Repeat 124 (blue lines). Stair-step complex CNVs (3 to 14 copies per genome) occur on Chr3L between two distinct long inverted repeat sequences: Repeat 124 (blue lines) and Repeat 127 (orange lines) (detailed in *Figure 3—figure supplement 1*). All four colonies are trisomic for the Chr3 centromere (*CEN3*, black circle) and all of Chr3R. (B) The $MIC_{50}$ increases with the copy number of the complex CNV. Heat map of isolate growth ($OD_{600}$ at 48 hr) in twofold increasing concentrations of miconazole. The $MIC_{50}$ is denoted with a yellow line. Each heat map represents the average of three independent $MIC_{50}$ assays (*Figure 2—source data 1*). (C) Schematic of the homologous chromosomes in AMS3051. The full length (gray) and dicentric CNV-containing (black) homologs with the positions indicated for *CEN3* (circle), the long repeat sequences (blue and orange lines), and SacII cut sites (dashed lines). Four regions (I–IV) that support a breakage-fusion-bridge mechanism for the formation of complex CNVs (see main text for details). (D) Allele ratio plot of all heterozygous loci located within and flanking the complex CNV for AMS3050 and AMS3051. Allele ratio plots for all isolates are in *Figure 3—figure supplement 2*. (E) SacII-digested CHEF karyotype of the progenitor and miconazole-evolved isolates (first lane of undigested AMS3050 is shown for relative size). The SacII digest isolates the region with variable copy number and *CEN3* (schematic at far right). CHEF gel stained with ethidium bromide (left panel) and analyzed by Southern blot using a DIG-labeled probe to *orf19.344*, located within the complex CNV (middle panel), and *CEN3* (right panel). Both Southern blot probes detect a novel band that increases in size as the complex CNV increases in copy number.

The online version of this article includes the following figure supplement(s) for figure 3:

**Figure supplement 1.** The Chr3L CNV is flanked by two, distinct inverted repeat sequences.

**Figure supplement 2.** CNVs are intra-chromosomal amplifications between two distinct inverted repeat sequences.

Chr3L. This region (as in *Figure 1*) was flanked by two distinct long inverted repeat sequences resulting in a stair-step amplification in AMS3051 and AMS3054 (Repeats 124 [blue lines] and Repeats 127 [orange lines], *Figure 3A*, *Figure 3—figure supplement 1*). In one isolate (AMS3052), the region of variable amplification extended beyond Repeat 127 to the centromere of Chr3 (*CEN3*). Importantly, the $MIC_{50}$ of the miconazole-evolved colonies increased as the maximum copy number of this complex CNV increased (*Figure 3B*), supporting that in an isogenic background the increase in copy number directly correlates with an increase in MIC.

The isogenic isolates contained five genomic features that provide clues concerning the mechanism of complex CNV formation on Chr3L (*Figure 3C*): I) Monosomy (and thus LOH) extending from Repeat 124 to the telomere; II) Complex CNVs with a maximum copy number that varies between the individual colonies and amplifies unique sequences between two distinct long inverted repeats (Repeats 124 and 127); III) Trisomy of sequences extending from Repeat 127 to *CEN3*; IV) Trisomy of *CEN3* and all of Chr3R; and V) Amplification of a single haplotype in the complex CNV of Chr3L and throughout the trisomic region of Chr3R (*Figure 3D*, *Figure 3—figure supplement 2*) (although this is not detectable in the region distal to Chr3R position 896,538, because this region is already homozygous in the progenitor). From these observations, we hypothesized that one homolog of Chr3 was intact and includes the monosomic portion of Chr3L, while the other homolog has formed a dicentric molecule that promotes breakage-fusion-bridge (BFB) cycles that result in the complex, stair-step amplifications between and within the long inverted repeats (*Figure 3C*).

If this hypothesis is correct, we should be able to detect dicentric chromosome intermediates by CHEF gel karyotype analysis. Because an intact dicentric Chr3 would be too large to resolve, we analyzed SacII-digested chromosomal DNA. SacII sites fall to the left of the complex CNV of Chr3L (within the monosomy) and to the right of *CEN3* yielding a fragment that should be ~480 kb in the AMS3050 progenitor (*Figure 3C and E*). Indeed, the progenitor and all four evolved isolates had a band of ~480 kb (*Figure 3E*). In addition, the four evolved isolates had an additional band of increased size (~854 kb to ~2.16 Mb) consistent with the level of amplification in these isolates. This high-molecular-weight band hybridized to two different probes: an ORF found within the most amplified region of the Chr3L CNV (*orf19.344*) and *CEN3*. Thus, the size and content of the SacII-digested region including *CEN3* is consistent with the idea that the fragment includes a dicentric chromosome that is linked via the region containing the different-sized complex CNVs found among the AMS3050 derivatives (*Figure 3C and E*).

## Recombination occurs between long inverted repeats leading to CNV formation

Models of BFB in both fungal and human cells support that dicentric chromosomes form via non-allelic homologous recombination (NAHR) between inverted repeat sequences (*Croll et al., 2013*; *Hermetz et al., 2014*; *Notta et al., 2016*). We used Oxford Nanopore Technology long-read sequencing to test the hypothesis that NAHR between repeat sequences was involved in the formation of the complex CNV on Chr3L in AMS3051 (*Figure 4A–C*) . Structural variants, indicators of recombination products not identified in the reference genome, were identified in AMS3051 using split-read alignments, read mismatching, and read depth analyses (see Materials and methods). One structural variant was detected at Repeat 124 (e.g. *Figure 4B*), and the other structural variant was detected at Repeat 127 (e.g. *Figure 4C*). Each structural variant combined unique (non-repeat) sequences, that were located up to ~100 kb apart in the reference genome, into a single long-read of only 8 kb – 10 kb in length. Approximately 50 unique long-reads supported these structural variants and each long-read included a single copy of either Repeat 124 or Repeat 127. The individual long-reads switched between the complement and reverse complement orientation (relative to the reference) within the long repeat sequence (Repeats 124 and 127, *Figure 4B and C*). These long-reads represent fold-back inversions that were presumably mediated by NAHR between distant copies of the long repeat sequences. These observations, together with the intra-chromosomal CNV expansions detected by CHEF (*Figure 3E*), suggest that the complex CNV located on Chr3L occurs via an accordion-like expansion.

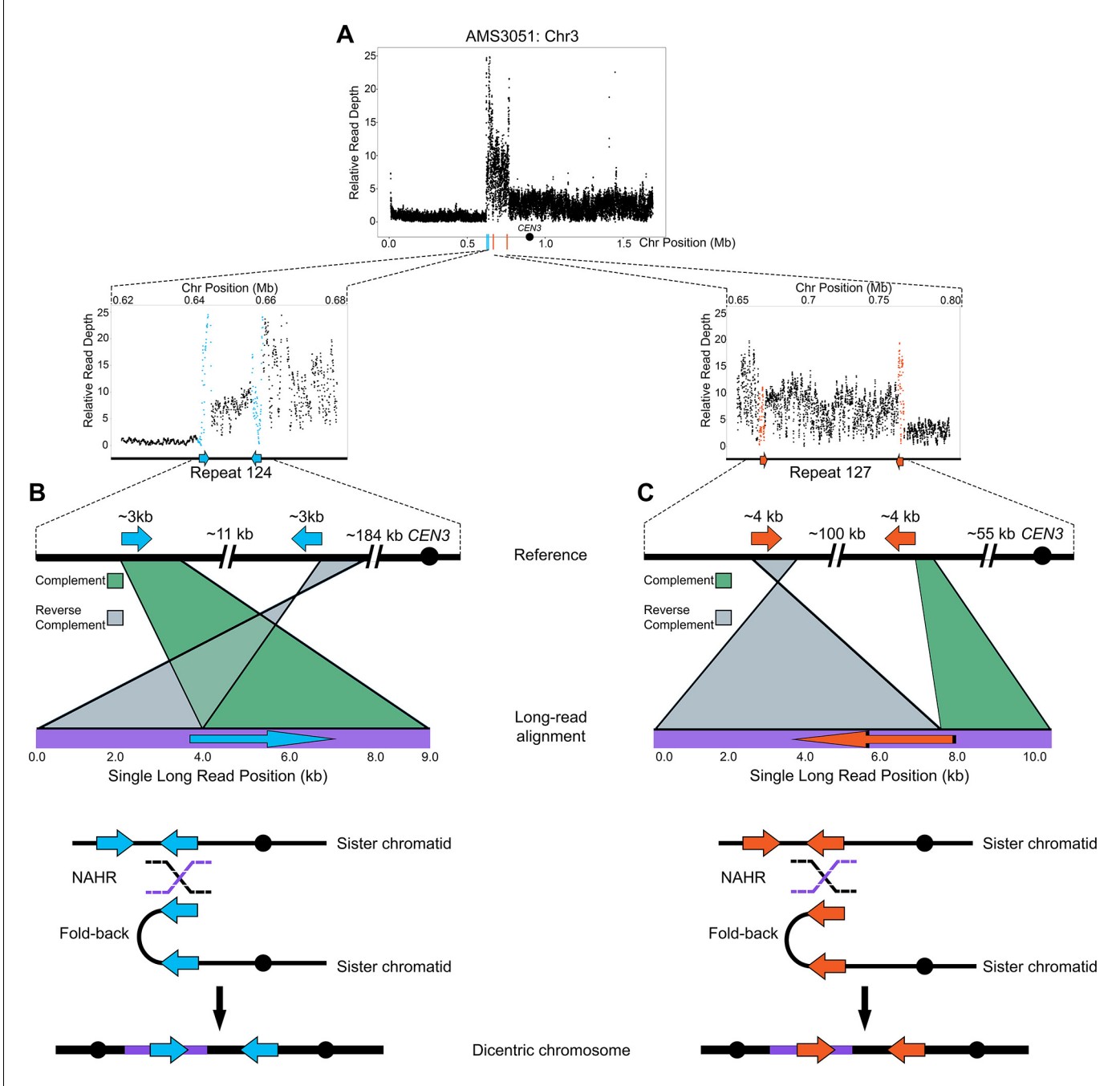

**Figure 4.** Long-read sequencing reveals recombination products involved in the formation of complex CNVs. (A) Relative Illumina read depth for the miconazole-evolved isolate AMS3051 plotted by chromosome position (data as in *Figure 3—figure supplement 1*) indicating the presence of two inverted repeats (Repeat 124 (blue line) and Repeat 127 (orange line)) flanking a complex CNV on Chr3L. Repeat numbers refer to *Supplementary file 2* from *Todd et al., 2019*. In the reference genome, Repeat 124 (B), consists of two inverted copies (~99% sequence identity) that are ~3 kb in length that are ~11 kb apart and Repeat 127 (C) consists of two inverted copies (~99% sequence identity) that are ~4 kb in length and located ~100 kb apart. Analysis of long-read sequences from AMS3051 identified structural variants relative to the reference genome at both Repeats 124 and 127. A single representative long-read (bottom purple line) aligned to the reference genome (top black line) is shown for Repeat 124 (B) and Repeat 127 (C). The long-read contains unique sequences that are separated by up to ~100 kb in the reference genome. Green-colored areas indicate alignment to the complement strand and gray-colored areas indicate alignment to the reverse complement strand of the reference genome. The transition between complement/reverse complement occurs within the repeat sequence. Schematics of both Repeat 124 and Repeat 127 indicate the formation of a fold-back inversion and non-allelic homologous recombination (NAHR, black and purple dashed lines) between repeat copies on sister chromatids, which could generate a dicentric chromosome. Alignment of the recombination product is inferred to produce the long-read that was detected (purple bar

*Figure 4 continued on next page*

*Figure 4 continued*

within the dicentric chromosomes). Each structural variant was supported by ~50 long-read sequences (see Materials and methods). All features of the CNV breakpoints are detailed in **Supplementary file 2**.

## Loss of the CNVs and subsequent LOH in the absence of antifungal drug selection

Highly amplified CNVs are expected to be subject to recombination events that reduce copy number, especially if selection for the extra gene copies is relaxed. To determine the stability of the Chr3L CNV in the absence of antifungal drug, we isolated single colonies on rich medium after 72 hr. Heterogeneous populations of large and small colonies were observed for each of the four miconazole-evolved isolates (*Figure 5A and B*). Both a large and small colony isolated from AMS3051 were plated for single colonies on rich medium: the large colony gave rise to similarly large colonies (AMS3092), while the small colony continued to give rise to a heterogeneous population of large (AMS3093) and small (AMS3094) colonies (*Figure 5A and B*). WGS and read depth analysis supported that the small colony phenotype in the absence of miconazole was due to a fitness defect associated with the dicentric chromosome and/or the monosomic portion of Chr3L. In contrast, both large colonies (AMS3092 and AMS3093) had resolved the dicentric chromosome and regained the disomic portion of Chr3L (*Figure 5A*, *Figure 5—figure supplement 1*). In one case (AMS3092), the dicentric chromosome underwent a recombination event that maintained the complex CNV and heterozygosity across *CEN3* and Chr3R. In the other case (AMS3093), the dicentric chromosome underwent a recombination event at *CEN3* that returned this isolate to a euploid genotype (*Figure 5—figure supplement 1*) and homozygosed all of Chr3L (*Figure 5—figure supplement 2*). In both examples, the dicentric chromosome was never completely lost, but had recombined to resolve the dicentric.

The impact of CNV loss on fitness was determined for the three single colonies derived from AMS3051 (*Figure 5C*). The highest $MIC_{50}$ (4 µg/ml miconazole) was observed for both isolates with the CNV on a dicentric chromosome (AMS3051 and AMS3094). Surprisingly, the $MIC_{50}$ decreased (2 µg/ml miconazole) for the isolate that retained the complex CNV (AMS3092) but had become euploid for all other regions on Chr3. One possibility is that, loss of the extra copy of Chr3R (which contains the multidrug resistance and tolerance genes described above: *MRR1*, *CDR1*, *CDR2,* and *CRZ1*) reduced the $MIC_{50}$ of this strain. Therefore, the decrease in $MIC_{50}$ (between AMS3051 and AMS3092) may be due to reduced copy number of these genes.

As expected, the $MIC_{50}$ was lowest (1 µg/ml miconazole) for the isolate that returned to the euploid genotype (AMS3093). This $MIC_{50}$ was still above the $MIC_{50}$ of the euploid progenitor AMS3050 (0.5 µg/ml miconazole) despite the lack of de novo SNVs between the two isolates (see Materials and methods). The difference in $MIC_{50}$ (between AMS3050 and AMS3093) may be due to the additional LOH of Chr3L that occurred during the resolution of the dicentric chromosome in AMS3093 (*Figure 5D*, *Figure 5—figure supplement 2*).

Finally, under constant antifungal drug selection (20 µg/ml miconazole), the dicentric chromosome appeared to be highly stable by both colony morphology and karyotype analysis (*Figure 5—figure supplement 3*). Thus, under continued antifungal drug selection the dicentric chromosome is maintained at the population and single-cell level. In contrast, removal from the selection pressure promotes the loss/resolution of the dicentric chromosome.

## Discussion

This study identifies a novel mechanism for generating beneficial CNVs during adaptation to physiologically relevant concentrations of azole antifungal drugs. The formation of these complex CNVs is rapid, expandable, and reversible. Recurrent CNV breakpoints occur in clinical isolates with diverse genetic backgrounds that are exposed to azole antifungal drugs and can increase fitness across multiple azole drugs. The heterozygous diploid genome of *C. albicans* has enabled us to determine the mechanism of CNV formation from population-level and single-cell karyotype analyses. Ultimately, the expansion and contraction of these CNVs affects the rate and dynamics in which antifungal drug resistance and tolerance is acquired, and provides a plausible explanation, independent of whole

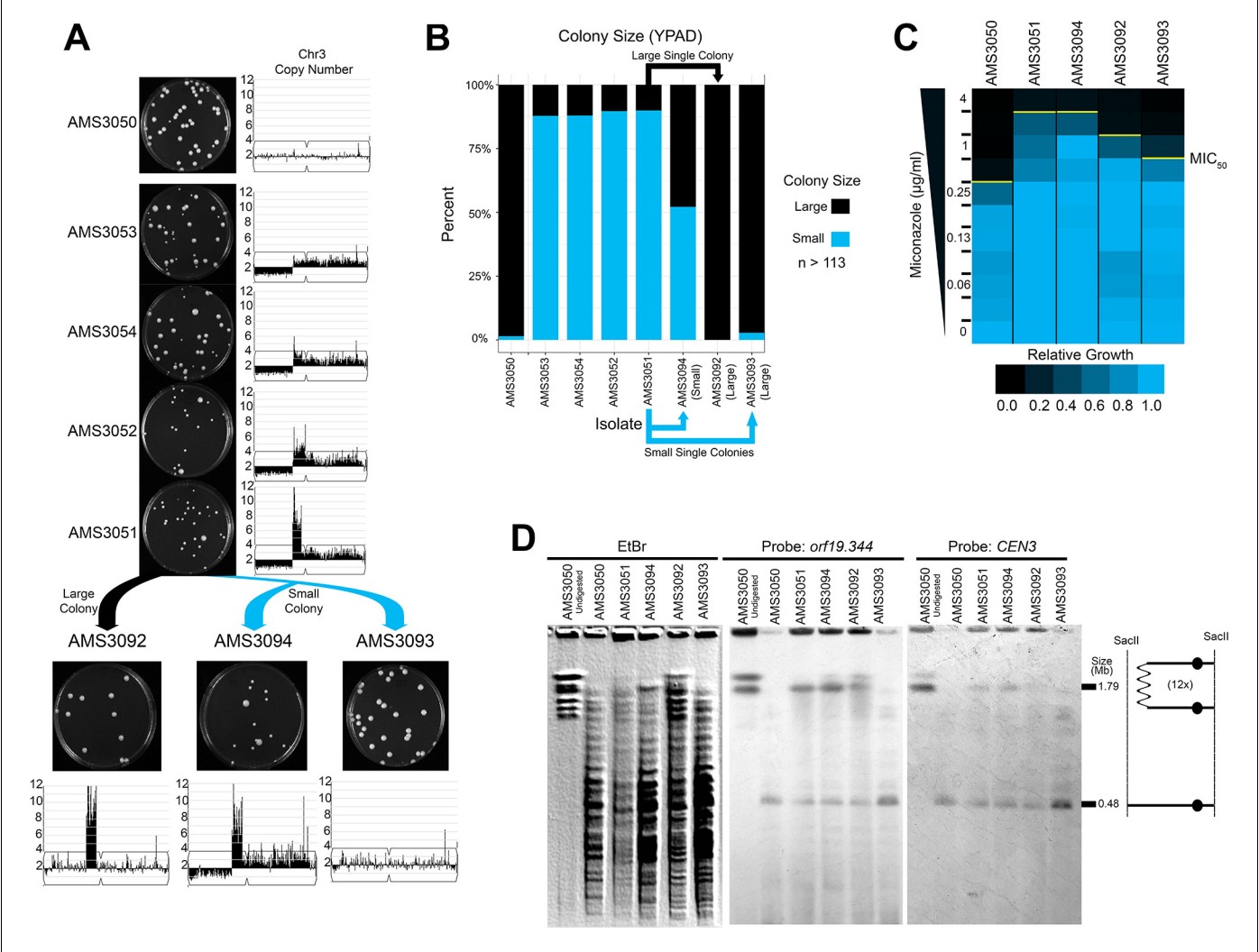

**Figure 5.** Complex CNVs resolve in the absence of antifungal drug by eliminating the dicentric chromosome. (**A**) Representative images of the progenitor (AMS3050) and the four miconazole-evolved isolates (AMS3053, AMS3054, AMS3052, and AMS3051) containing the Chr3L CNV grown on YPAD. Copy number of Chr3 (from *Figure 3*) shown to the right of the plate images. The notch in Chr3 is *CEN3*. Representative images below AMS3051 of single colonies derived from either a small (blue) or large (black) colony of AMS3051 on rich medium: The small colony gave rise to AMS3094 and AMS3093 and the large colony gave rise to AMS3092. Copy number of Chr3 shown below the plate images. Whole genome sequencing data are provided in *Figure 5—figure supplement 1*. (**B**) Colony size analysis using ImageJ (see Materials and methods), n > 113 (up to n = 300) single colonies, three biological replicates. (**C**) Heat map of $OD_{600}$ values taken at 48 hr in twofold increasing concentrations of miconazole. The $MIC_{50}$ is denoted with a yellow line. Each heat map represents the average of three independent $MIC_{50}$ assays (*Figure 2—source data 1*). (**D**) CHEF of the parental strain and miconazole-evolved isolates. Whole genomic DNA was digested with SacII to isolate the region containing the CNV and *CEN3* (schematic to right). CHEF gel stained with ethidium bromide (left panel) and analyzed by Southern blot with DIG-labeled probes to *orf19.344* within the CNV (middle panel) and *CEN3* (right panel).

The online version of this article includes the following figure supplement(s) for figure 5:

**Figure supplement 1.** Loss of Chr3 CNV correlates with reduction of $MIC_{50}$ to miconazole.
**Figure supplement 2.** Additional loss of heterozygosity can occur during resolution of complex CNVs.
**Figure supplement 3.** Dicentric Chr3 is stable in the presence of antifungal drug.

chromosome aneuploidy, for the phenotypic heterogeneity frequently reported during antifungal drug susceptibility testing (e.g., tolerance or trailing growth and heteroresistance) (*Ben-Ami et al., 2016*; *Qiao et al., 2017*; *Sionov et al., 2009*).

## Model for complex CNV formation

The complex CNVs amplified recurrent chromosome regions and were associated with increased fitness in azole antifungals. Short-read and long-read whole genome sequencing revealed that all of the complex CNVs have copy number breakpoints that occur within distinct long inverted repeat sequences; in other words, the amplified regions reside between sets of different inverted repeat sequences. Based on these breakpoint sequences, we propose a model of NAHR and breakage-fusion-bridge (BFB) cycles for the formation and resolution of complex CNVs in *C. albicans* (*Figure 6*). For example, to generate the Chr3L complex CNV, a DNA double-strand break (DSB) occurs (the exact location of the DSB is not known) between the telomere and the telomere-proximal inverted repeat (*Figure 6A and B*, Repeat 124, blue arrows). This DSB generates a telomere proximal acentric fragment that is lost during future cell divisions. After DNA replication, NAHR between the long inverted repeat sequences (*Figure 6A and B*, Repeat 124, blue arrows) located on sister chromatids generates a dicentric chromosome and amplification of all sequence from the site of NAHR to the telomere on Chr3R (*Croll et al., 2013*; *Lang et al., 2013*; *Ramakrishnan et al., 2018*; *Stimpson et al., 2012*). Alternatively, the dicentric chromosome could be formed via an intra-chromosomal fold-back-like mechanism between repeat copies that primes break induced replication (BIR) (*Narayanan et al., 2006*; *Rattray et al., 2005*). During cytokinesis, the dicentric chromosome bridge is broken preferentially at a centromere-proximal location due to closure of the actomyosin ring (*Lopez et al., 2015*); the longer, monocentric chromosome fragment is followed in the model. During the next cell cycle, DNA replication generates two sister chromatids that again undergo NAHR, this time between the two copies of the centromere-proximal long inverted repeat sequence (*Figure 6A and B*, Repeat 127, orange arrows), which generates a second dicentric chromosome and an amplification of all sequence from the site of the second NAHR event to the telomere on Chr3R. These dicentric BFB cycles can result in heterogeneous outcomes that are beneficial in the presence or absence of antifungal drug selection including: further amplification of the repeat and unique sequences (*Figure 6C*); NAHR with a repeat on the other homolog, resulting in resolution of the dicentric and isolation of the CNV on a monocentric chromosome (*Figure 6D*, top); and recombination (likely BIR) that occurs with the other homolog at a position centromere proximal to the CNV and continues to the Chr3L telomere, resulting in loss of the complex CNV and homozygosis of Chr3L (*Figure 6D*, bottom). Example symmetric stair-step copy number amplifications generated by these recombination events and the presence of fold-back inversions that arise during NAHR between repeat sequences are proposed in the model (*Figure 6C and D*). We propose a similar model for asymmetric stair-step copy number amplifications that can form due to recombination events involving three (instead of two) distinct inverted repeat sequences (*Figure 6—figure supplement 1*). Finally, the Chr3L CNV expansions (*Figure 3*, during antifungal selection) and contractions (*Figure 5*, when selection is relaxed) were detected after formation of a single colony and underscore the dynamic potential of these CNVs. Accordingly, we propose that a dicentric chromosome intermediate is driving the rapid generation of copy number amplifications and sub-clonal heterogeneity observed in *C. albicans* clinical isolates.

A limitation of this study is that the rate of complex CNV formation is not known, and it is also unclear whether these CNVs occur in the context of a host infection. Dicentric chromosomes have been identified in FLC-resistant clinical isolates of *C. albicans* (*Selmecki et al., 2006*), and suggest that BFB cycles may be possible in vivo. Additionally, CNVs with breakpoints at long-repeat sequences have been identified after passage of *C. albicans* through a mouse model of oropharyngeal candidiasis (in the absence of antifungal drugs) (*Todd et al., 2019*), and suggest that complex CNVs could occur in vivo. Importantly, there was very little fitness cost associated with maintaining a complex CNV on a monocentric chromosome (*Figure 2*), which underscores that these CNVs are well-tolerated in *C. albicans*. Ultimately, future studies using single-cell analyses (e.g. *Lauer et al., 2018*) are needed to determine the rate of complex CNV formation both in vitro and in vivo in the presence and absence of antifungal drugs.

Stair-step amplifications and fold-back inversions generated via BFB cycles are also observed in human cancers and developmental diseases (*Cheng et al., 2016*; *Hermetz et al., 2014*; *Marotta et al., 2017*). For example, amplification of oncogenes (*EFGR, ERBB2, and MYC*) occurs via BFB in up to ~70% of diverse tumor types (including breast, colorectal, lung, and liver cancers) (*Marotta et al., 2017*; *Venkataram et al., 2016*). While the exact molecular mechanisms of DNA

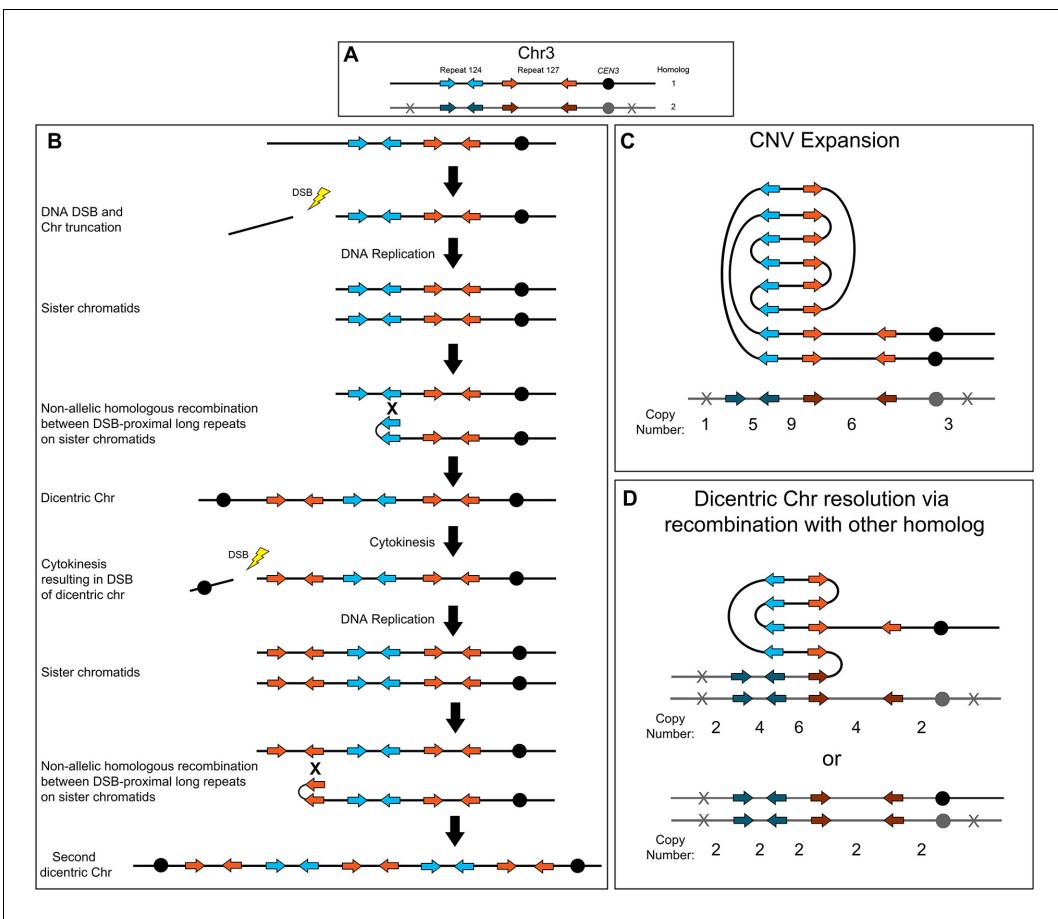

**Figure 6.** Breakage-fusion-bridge model for the formation and resolution of complex CNVs. DNA double-strand breaks (DSBs), recombination, and dicentric chromosome formation can drive complex CNV formation through successive breakage-fusion-bridge (BFB) cycles. (**A**) Two distinct long-repeat sequences, Repeat 124 (blue/dark blue arrows) and Repeat 127 (orange/dark orange arrows), are located on both Chr3 homologs (black and gray). Heterozygous SNVs indicated with an X on Chr3 homolog 2. (**B**) A DSB occurs telomere proximal to the left copy of Repeat 124 on homolog 1 of Chr3, resulting in the formation of a telomere-proximal acentric DNA fragment. DNA replication generates two sister chromatids with a truncated arm. Non-allelic homologous recombination (NAHR) between Repeat 124 on sister chromatids generates a dicentric chromosome. During cytokinesis, the dicentric chromosome undergoes bridge formation and breaks near the centromere due to the closure of the actomyosin ring, generating two asymmetric chromosome fragments, each with only one centromere. The longer, monocentric chromosome fragment is followed in this model. During the next cell cycle, DNA replication generates two sister chromatids that undergo NAHR between a distinct long inverted repeat (Repeat 127, orange arrows) on sister chromatids that generates a second dicentric chromosome. Subsequent BFB cycles can result in different outcomes depending on the environmental selection: (**C**) Complex CNV expansion in which successive rounds of the BFB cycle generate CNVs with higher copy numbers, or (**D**) NAHR with a repeat on the other homolog results in the resolution of the dicentric chromosome and isolates the CNV on a monocentric chromosome (top, e.g. AMS3092). Alternatively, break induced replication (BIR) could prime near *CEN3* and continue to the Chr3L telomere, resulting in the loss of the complex CNV and homozygosis of Chr3L (bottom, e.g. AMS3093). Example stair-step copy numbers of genomic segments indicated below each schematic.

The online version of this article includes the following figure supplement(s) for figure 6:

**Figure supplement 1.** Breakage-fusion-bridge model for the formation of asymmetric complex CNVs.

repair during BFB in human cells remain under investigation (*Cheng et al., 2016*; *Maciejowski et al., 2015*; *Marotta et al., 2012*; *Tanaka et al., 2007*), fold-back inversions causing CNVs occur more frequently (13/21 vs. 3/21) at breakpoints of microhomology (2–8 bp) than at regions of longer homology (296–330 bp, 82–95%), such as LINE, SINE, and Alu elements that are present in thousands of copies per genome (*Hermetz et al., 2014*; *Rodić and Burns, 2013*). In

comparison, the copy number breakpoints identified in *C. albicans* only occurred at long-repeat sequences (median 1513 bp) that shared high sequence identity (median 96.1%) that were predominantly present (10/12 repeats) in only two copies per genome and were always located on the same chromosome arm. Therefore, while *C. albicans* appears to have a strong preference for a BFB repair mechanism that involves homologous recombination within long-repeat sequences, the relatively low copy number of these repeat sequences in the *C. albicans* genome has enabled more precise mapping of the CNV breakpoints and fold-back inversions, whereas similar events remain challenging to resolve in the human genome.

## Impact of CNVs on the mutational landscape

CNVs can dramatically alter population dynamics and mutational landscapes. The rate with which CNVs occur (~$1.3\times10^{-6}$ to ~$1.5\times10^{-6}$ per gene per cell division) is several orders of magnitude higher than the rate of SNVs (~$0.33\times10^{-9}$ per site per cell division) in the absence of selection (*Lynch et al., 2008*). In the presence of strong selection, for example nutrient limitation, the rate of CNV formation can be much higher and can rapidly drive clonal interference between unique CNV-containing lineages within the population (*Lauer et al., 2018*; *Payen et al., 2014*). Our single colony analyses support that formation of a dicentric chromosome can generate continuous cycles of genome instability and can increase population heterogeneity, which is likely to contribute to high rates of clonal interference.

We hypothesize that the complex CNVs identified here were selected due to the presence of genes that, when amplified, provide a fitness benefit in the presence of azole antifungal drugs (*Selmecki et al., 2008*). In support of this hypothesis, *NPR2* (within the Chr3L CNV) results in increased azole resistance similar to acquisition of the CNV (*Mount et al., 2018*). Other genes like *ERG11* and *TAC1* (both within the i(5L) CNV) can confer copy number dependent increases in azole resistance (*Selmecki et al., 2008*).

In addition to the direct effect of gene amplification (i.e. more copies of the gene and its products), every cell that acquires a complex CNV also has an increased target size for the acquisition of rare, single nucleotide variants (*Cone et al., 2017*; *Elde et al., 2012*). CNVs therefore provide a potential for an increased likelihood of beneficial, drug-resistant point mutations during antifungal drug selection. This provides a good explanation for the utility of CNVs and aneuploidy as intermediates in the acquisition of more stable mutations (e.g. single nucleotide variants) (*Elde et al., 2012*; *Ford et al., 2015*; *Roth and Andersson, 2004*; *Sun et al., 2009*; *Yona et al., 2012*).

Importantly, many of the genes that are known to cause azole resistance contain SNVs that are recurrently found in drug-resistant isolates. For example, recurrent SNVs in *MRR1*, *ERG11*, *FUR1*, and *FKS1* are found in drug-resistant isolates of *C. albicans* and interestingly, are recurrent in distantly related *Candida* species as well, including *C. glabrata* and *C. auris* (*Flowers et al., 2015*; *Garcia-Effron et al., 2009*; *Hope et al., 2004*; *Morschhäuser et al., 2007*; *Muñoz et al., 2018*; *Perlin, 2011*). Both *MRR1* (Chr3R) and *ERG11* (Chr5L), genes with recurrent SNVs known to cause azole resistance, were amplified in CNVs in this study, supporting the idea these CNVs can amplify genes that confer increased fitness in the presence of antifungal drugs and simultaneously increase the probability that a *bona fide* drug resistance allele will occur in those same genes.

LOH is also an important mechanism of acquired antifungal drug resistance, and exposure to azole antifungal drugs can increase the frequency of LOH in *C. albicans* (*Bouchonville et al., 2009*; *Dunkel et al., 2008*; *Forche et al., 2011*; *Niimi et al., 2010*). Heterozygous mutations in both *MRR1* and *FKS1* also can undergo LOH in the presence of antifungal drug selection for homozygosis of the beneficial allele (*Dunkel et al., 2008*; *Niimi et al., 2010*). Here, we obtained direct evidence that LOH can arise via dicentric chromosome formation and telomere-proximal chromosome loss, the same mechanism that yields complex CNVs. For example, new regions of LOH were identified during dicentric chromosome resolution (e.g. AMS4702 in *Figure 1—figure supplement 3*, and AMS3093 in *Figure 5—figure supplement 2*). These data further support that the long repeat sequences associated with CNV and LOH breakpoints are a major source of genome plasticity in the presence and absence of antifungal drug (*Todd et al., 2019*), and may underlie the variable mutation rates (hotspots) observed across an individual chromosome in other fungi as well (*Lang and Murray, 2011*).

Finally, we found that complex CNVs frequently occurred in regions of the genome that were already homozygous in the euploid progenitors (*Figure 1—figure supplement 3*). We propose that

such homozygous regions are the result of prior rounds of complex CNV formation and subsequent LOH in some of these isolates. Other regions of long-track homozygous sequence are frequently observed in diverse clinical isolates of *C. albicans* (*Ene et al., 2018*; *Ford et al., 2015*; *Hirakawa et al., 2015*; *Ropars et al., 2018*; *Todd et al., 2019*) and may provide evidence that other complex CNVs resulting in LOH are occurring in some of these isolates as well, supporting the idea that these CNVs are transient in nature.

## CNVs promote antifungal drug tolerance

Antifungal drug tolerance is the ability of a subpopulation of cells within a susceptible isolate to grow slowly at drug concentrations above the $MIC_{50}$ (*Berman and Krysan, 2020*; *Rosenberg et al., 2018*). Mechanisms that contribute to the tolerance phenotype remain to be identified; however, genes involved in core stress pathways and cell wall/cell membrane biosynthesis appear to have an important role (*Berman and Krysan, 2020*; *Cowen et al., 2015*; *Rosenberg et al., 2018*). The complex CNVs detected here, together with what is known about drug responses in general, make it tempting to speculate on the specific genes that may be involved in the tolerance phenotype. These include genes encoding proteins involved in stress responses (*CRZ1*, *HSP70*, *CGR1*, *ERO1*, *TPK1*, *ASR1*, and *PBS2*) and cell wall/cell membrane integrity (*CDR3*, *NCP1*, *ECM21*, *MNN23*, *RHB1* and *KRE6*). We propose that amplification of these genes within transient CNVs is one major route to increase cellular fitness and generate the population heterogeneity that underlies antifungal drug tolerance.

## Impact of azole antifungals on centromere function and CNV formation

Whole chromosome missegregation resulting in aneuploidy is well-documented in *C. albicans* and other fungal pathogens (*Forche et al., 2009*; *Forche et al., 2019*; *Gerstein et al., 2015*; *Janbon et al., 1998*; *Ngamskulrungroj et al., 2012*; *Poláková et al., 2009*; *Reedy et al., 2009*; *Rustchenko-Bulgac, 1991*; *Selmecki et al., 2006*; *Selmecki et al., 2009*; *Sionov et al., 2013*; *Yang et al., 2019*). Many of these aneuploidies provide a selective benefit in vitro and in vivo, and in the presence and absence of antifungal drugs. Importantly, whole chromosome aneuploidy is likely to be induced (as well as selected) by drug exposure (*Harrison et al., 2014*); however, the mechanisms driving chromosome mis-segregation are not well characterized.

We find that the acquisition of complex CNVs involves the formation of a dicentric chromosome intermediate. The dicentric chromosomes are maintained in the presence of drug (*Figure 5—figure supplement 3*), whereas elimination of drug selection results in a ~13% increase in colonies that have lost the dicentric chromosome via subsequent recombination events that either isolate the complex CNV on a chromosome arm or revert to the euploid progenitor genotype (*Figure 5A and B*). Stabilization of dicentric chromosomes has been observed in humans, *Drosophila melanogaster*, *Zea mays*, and *Schizosaccharomyces pombe* due to centromere inactivation (*Agudo et al., 2000*; *Earnshaw and Migeon, 1985*; *Han et al., 2006*; *Sato et al., 2012*; *Stimpson et al., 2012*; *Sullivan and Schwartz, 1995*). Therefore, in addition to selection, we propose that the dicentric chromosome can be stabilized in the presence of azole drugs due to the depletion of Cse4/CENP-A, the centromere-specific histone H3. Importantly, Cse4/CENP-A, which is normally enriched at centromeric DNA, is depleted from the centromeres of *C. albicans* cells exposed to FLC (10 µg/ml), and contributes to an increased rate of chromosome mis-segregation (*Brimacombe et al., 2019*). Whether Cse4/CENP-A is actively removed from centromeric DNA in the presence of azole drugs is not known, but mammalian models suggest that CENP-A can be recruited away from the centromere to sites of DNA damage (*Zeitlin et al., 2009*). The presence of azole drugs may increase the likelihood that a dicentric chromosome is maintained and further increase the likelihood of recombination events that result in complex CNV formation. Therefore, determining the mechanistic link between antifungal drug treatment and centromere function is critical to understanding the genome instability that occurs during acquisition of drug resistance, including whole-chromosome aneuploidy, dicentric chromosome intermediates, and complex CNVs.

## Conclusion

Complex CNVs are acquired rapidly during adaptation to azole antifungal drugs and cause an increase in drug resistance and tolerance. These CNVs are found across the genome, occur in

diverse genetic backgrounds, and are all formed between a set of two distinct inverted repeat sequences that flank the amplified region. Evidence here provides support for a mechanism of CNV generation and resolution back to euploidy that is driven by successive BFB cycles involving a dicentric chromosome that repairs via homologous recombination between the repeat sequences. Furthermore, the cell-to-cell variability observed for clinical isolates during drug susceptibility assays may be due to the heterogeneity in the copy number of CNVs present within individual cells in a population as well as continued BFB cycles. Identification of CNVs in other pathogenic fungi, including the emerging multi-drug-resistant pathogen *C. auris,* further suggests that this mechanism of CNV formation may occur in diverse species. Together, these findings suggest a novel mechanism for transient CNV formation that increases the adaptive potential of fungal pathogens to antifungal drugs.

# Materials and methods

### Key resources table

| Reagent type (species) or resource | Designation | Source or reference | Identifiers | Additional information |
|---|---|---|---|---|
| Strain, strain background (*Candida albicans*) | SC5314 | *Hirakawa et al., 2015* (DOI: 10.1101/gr.174623.114) | RRID:SCR_013437 | |
| Strain, strain background (*C. albicans*) | P75016 | *Hirakawa et al., 2015* (DOI: 10.1101/gr.174623.114) | | |
| Strain, strain background (*C. albicans*) | P75063 | *Hirakawa et al., 2015* (DOI: 10.1101/gr.174623.114) | | |
| Strain, strain background (*C. albicans*) | P78042 | *Hirakawa et al., 2015* (DOI: 10.1101/gr.174623.114) | | |
| Antibody | Anti-Digoxigenin-AP Fab Fragments (Polyclonal from sheep) | Roche | 11093274910 RRID:AB_2734716 | (1:5000) |
| Sequence-based reagent | PCR Primers | This study | | *Supplementary file 5* |
| Commercial assay or kit | Illumina Nextera XT Library Kit | Illumina | 105032350 | |
| Commercial assay or kit | Illumina Nextera XT Index Kit | Illumina | 105055294 | |
| Commercial assay or kit | Illumina Nextera Flex DNA Kit | Illumina | 20018704 | |
| Commercial assay or kit | Illumina Nextera DNA CN Index kit | Illumina | 20018707 | |
| Commercial assay or kit | Blue Pippin 1.5% agarose gel dye-free cassette | Sage Science | 250 bp - 1.5 kb DNA size range collections, Marker R2 | Target of 900 bp |
| Commercial assay or kit | Illumina MiSeq v2 Reagent Kit | Illumina | 15033625 | 2 × 250 bp |
| Commercial assay or kit | 1D Ligation Sequencing Kit | Oxford Nanopore Technologies | SQK-LSK108 | |
| Commercial assay or kit | R9 FLO-Min106 spot-on flow cell | Oxford Nanopore Technologies | R9.4.1 | |
| Commercial assay or kit | Ultra II End Repair/dA-Tailing Module | New England Biolabs | E7546S | |

*Continued on next page*

*Continued*

| Reagent type (species) or resource | Designation | Source or reference | Identifiers | Additional information |
|---|---|---|---|---|
| Commercial assay or kit | Qubit dsDNA HS kit | Life Technologies | Q32854 | |
| Commercial assay or kit | PCR DIG Probe Synthesis Kit | Roche | 11636090910 | |
| Commercial assay or kit | Agilent 2100 Bioanalyzer High Sensitivity DNA Reagents | Agilent Technologies | 5067–4626 | |
| Commercial assay or kit | SacII restriction enzyme | New England Biolabs | R0157S | |
| Chemical compound, drug | Fluconazole (FLC) | Alfa Aesar | J62015 | |
| Chemical compound, drug | Miconazole | Alfa Aesar | AAJ6087206 | |
| Chemical compound, drug | Itraconazole | Alfa Aesar | AAJ6639003 | |
| Chemical compound, drug | Posaconazole | MilliporeSigma | 11-101-3331 | |
| Chemical compound, drug | Ketoconazole | Fisher Scientific | AC455470010 | |
| Chemical compound, drug | PMSF | Milipore Sigma | 10837091001 | |
| Software, algorithm | Trimmomatic | *Bolger et al., 2014* (DOI: 10.1093/bioinformatics/btu170) | v0.33 RRID:SCR_011848 | |
| Software, algorithm | BWA | *Li, 2013* (DOI: 10.1093/bioinformatics/btp324) | v0.7.12 RRID:SCR_010910 | |
| Software, algorithm | Samtools | *Li et al., 2009* (DOI: 10.1093/bioinformatics/btp352) | v0.1.19 RRID:SCR_002105 | |
| Software, algorithm | Genome Analysis Toolkit | *McKenna et al., 2010* (DOI: 10.1101/gr.107524.110) | v3.4–46 RRID:SCR_001876 | |
| Software, algorithm | Yeast Analysis Mapping Pipeline | *Abbey et al., 2014* (DOI: 10.1186/s13073-014-0100-8) | V1.0 | |
| Software, algorithm | JMP Pro | https://www.jmp.com | V14.2.0 | |
| Software, algorithm | ImageJ | https://imagej.nih.gov/ij/? | v2.0.0-rc-30/ 1.49 s RRID:SCR_003070 | |
| Software, algorithm | Integrative Genomics Viewer | *Thorvaldsdóttir et al., 2013* (DOI: 10.1093/bib/bbs017) | v2.3.92 RRID:SCR_011793 | |
| Software, algorithm | R | https://www.r-project.org | v3.5.2 RRID:SCR_001905 | |
| Software, algorithm | Candida Genome Database | http://Candidagenome.org | RRID:SCR_002036 | |
| Software, algorithm | NGMLR | *Sedlazeck et al., 2018* (DOI: 10.1038/s41592-018-0001-7) | V0.2.7 | |
| Software, algorithm | Sniffles | *Sedlazeck et al., 2018* (DOI: 10.1038/s41592-018-0001-7) | v1.0.11 | |

*Continued*

| Reagent type (species) or resource | Designation | Source or reference | Identifiers | Additional information |
|---|---|---|---|---|
| Software, algorithm | SplitThreader | *Nattestad et al., 2016* (DOI: 10.1101/087981) http://splitthreader.com | | |
| Software, algorithm | Ribbon | http://genomeribbon.com | | |

## Yeast isolates and culture conditions

All isolates used in this study are described in *Supplementary file 1*. Isolates were stored at −80°C in 20% glycerol. Isolates were cultured at 30°C in YPAD medium (yeast extract, peptone, and 2% dextrose) supplemented with 40 μg/ml adenine and 80 μg/ml uridine. For *Figure 5*, isolates (AMS3050, AMS3053, AMS3054, AMS3052, and AMS3051) were grown at 30°C on YPAD agar plates (yeast extract, peptone, 2% dextrose, and 2% agar) for 48 hr. One single large and one single small colony (See ImageJ colony size analysis in Materials and methods) were selected and were re-plated for single colonies on YPAD agar plates for 48 hr at 30°C. From the large colony, a single large colony (AMS3092) was selected for WGS, antifungal drug susceptibility assays (MIC$_{50}$), colony size analysis, and CHEF analysis. From the single small colony, both a small (AMS3094) and large (AMS3093) colony were selected for WGS, antifungal drug susceptibility assays (MIC$_{50}$), colony size analysis, and CHEF analysis.

## In vitro evolution experiment

FLC susceptible progenitor isolates were plated for single colonies onto YPAD medium and incubated for 48 hr at 30°C. Twelve single colonies were isolated from each progenitor (SC5314, P75016, P75063, and P78042) and grown to stationary phase in 5 ml liquid YPAD to generate 48 independent lineages. A 1:1000 cell dilution was made in YPAD medium containing 1 μg/ml FLC in deep-well 96-well plates. Plates were sealed with Breathe EASIER tape (Electron Microscopy Sciences) and placed in a humidified chamber for 72 hr at 30°C. Every 72 hr, cells were resuspended and transferred into fresh medium containing 1 μg/ml FLC to a final cell dilution of 1:1000. In total, 10 transfers were conducted. After the final transfer, cells were collected for storage at −80°C, genomic DNA isolation, and MIC analysis.

## Microdilution minimum inhibitory concentration (MIC)

The MIC$_{50}$ for each isolate was measured using a microwell broth dilution. Isolates were inoculated from frozen stocks into YPAD medium and grown for 16 hr at 30°C. Cells were diluted in fresh YPAD medium to a final OD$_{600}$ of 0.01, and 10 μl of this dilution was inoculated into a 96-well plate containing 190 μl of a 0.5X dextrose YPAD medium with a twofold serial dilution of the antifungal drug or a no-drug control. Cells were incubated at 30°C in a humidified chamber and OD$_{600}$ readings were taken at both 24 and 48 hr post inoculation. The MIC$_{50}$ of each of the isolates was determined as the concentration of antifungal drug that decreased the OD$_{600}$ by ≥50% of the no-drug control. Supra-MIC Growth (SMG) was calculated by taking the average 48 hr growth of the wells above the 24 hr MIC$_{50}$ and dividing by the control well containing no drug (*Rosenberg et al., 2018*).

## Growth curve analysis

Isolates were inoculated from frozen stocks into YPAD medium and grown for 16 hr at 30°C. Cells were diluted in fresh YPAD medium to a final OD$_{600}$ of 0.01, and 10 μl of this dilution was inoculated into a 96-well plate containing 190 μl of a 1x dextrose YPAD medium with or without 1 μg/ml FLC. Cells were grown at 30°C in the BioTek Epoch with dual-orbital shaking (256 rpm) for 36 hr. OD$_{600}$ readings were taken every 15 min and plotted in R (v3.5.2) using ggplot2. Each growth curve was conducted in biological triplicate in three separate experiments. Calculation of summary statistics of each growth curve was conducted using the R package Growthcurver using standard parameters (*Supplementary file 3*; *Sprouffske and Wagner, 2016*).

## Contour-clamped homogenous electric field (CHEF) electrophoresis

Sample plugs were prepared as previously described (*Selmecki et al., 2005*). Briefly, cells were suspended in 300 µL 1.5% low-melt agarose (Bio-Rad) and digested with 1.2 mg Zymolyase (US Biological) at 37°C for 16 hr. Plugs were washed twice in 50 mM EDTA and treated with 0.2 mg/ml proteinase K (Alpha Azar) at 50°C for 48 hr. For samples digested with SacII, plugs were washed twice with 1x TE and incubated in 1 mM PMSF (Milipore Sigma) at room temperature for 30 min, washed twice with 1X TE and suspended in 1X CutSmart Buffer (New England Biolabs), and digested with 30 units of SacII (New England Biolabs) at 37°C for 24 hr. Chromosomes were separated in a 1% Megabase agarose gel (BioRad) in 0.5X TBE using the CHEF DRIII Pulsed Field Electrophoresis System. For whole chromosome separation, run conditions as follows: 60 s to 120 s switch, 6 V/cm, 120° angle for 36 hr followed by 120 s to 300 s switch, 4.5 V/cm, 120° angle for 12 hr. SacII digested chromosomes were separated as follows: 7 s to 100 s switch, 4.5 V/cm, 120° angle for 21 hr followed by 80 s to 400 s switch, 3.5 V/cm, 120° angle for 21 hr. CHEF gels were stained with ethidium bromide and imaged with the GelDock XR Imaging system (BioRad).

## Southern blot hybridization

Chromosomes from CHEF gels were transferred to a BrightStar Plus nylon membrane (Invitrogen). Hybridization and detection of the DNA was conducted as previously described (*Selmecki et al., 2005*; *Selmecki et al., 2008*; *Selmecki et al., 2009*; *Todd et al., 2019*). Probes were generated through PCR incorporation of DIG-11-dUTP into target sequences following the manufacturer's instructions (Roche). Primers used in this study are located in *Supplementary file 5*.

## Gene Ontology (GO) analysis

GO analysis was conducted for all terms (process, function, and component) using the GO Term Finder from the Candida Genome Database (CGD accessed 03/03/2020, http://www.candidagenome.org/cgi-bin/GO/goTermFinder). All genes located within the complex CNVs were included in the analysis. Genes located in other aneuploid chromosomes (AMS4104 - Chr7; AMS4106 - Chr3, i (5L); AMS4444 - Chr3, Chr4R, Chr7R) were not included in the GO analysis. Terms were considered significantly enriched if p<0.05. GO enrichment was determined for all genes included in the complex CNVs, as well as for each CNV individually (*Supplementary file 4*).

## ImageJ colony size analysis

All agar plates were imaged using the GelDock XR Imaging system (Bio-rad) using the same zoom and focal length. Images were exported as .png files, converted to eight-bit, and analyzed with Fiji (v2.0.0-rc-30/1.49 s) (*Schindelin et al., 2012*). An automatic threshold was set using the RenyiEntropy algorithm and the area of each particle was measured (*Sezgin and Sankur, 2004*). Colonies were considered small if their total area was less than two standard deviations below the mean colony size of the progenitor isolate, AMS3050 in the absence of miconazole. Colony size of each isolate, in the absence or presence of 20 µg/ml miconazole, was obtained from three individual agar plates (n > 113).

## Illumina whole genome sequencing

Genomic DNA was isolated using a phenol-chloroform extraction as described previously (*Selmecki et al., 2006*). Libraries were prepared using either the Illumina Nextera XT DNA Library Preparation Kit or the Nextera DNA Flex Library Preparation Kit. Adaptor sequences and low-quality reads were trimmed using Trimmomatic (v0.33 LEADING:3 TRAILING:3 SLIDINGWINDOW:4:15 MINLEN:36 TOPHRED33) (*Bolger et al., 2014*). Trimmed reads were mapped to the *C. albicans* reference genome (A21-s02-m09-r08) from the *Candida* Genome Database (http://www.candidagenome.org/download/sequence/C_albicans_SC5314/Assembly21/archive/C_albicans_SC5314_version_A21-s02-m09-r08_chromosomes.fasta.gz). Reads were mapped using BWA-MEM (v0.7.12) with default parameters (*Li, 2013*). PCR duplicated reads were removed using Samtools (v0.1.19) (*Li et al., 2009*), and realigned around predicted indels using the Genome Analysis Toolkit (RealignerTargetCreator and IndelRealigner, v3.4–46) (*McKenna et al., 2010*). All Illumina data have been deposited in the National Center for Biotechnology Information Sequence Read Archive database under PRJNA613282.

## Visualization of aneuploid chromosomes

Aneuploid chromosomes were visualized using the Yeast Analysis Mapping Pipeline (YMAP v1.0) (*Abbey et al., 2014*). Fastq files were uploaded to YMAP and read depth was determined and plotted as a function of chromosome position. Read depth was corrected for both GC-content and chromosome-end bias.

## Read depth analysis plots

For each isolate, read depth for each position within the genome was calculated using samtools depth (`-aa`) (v0.1.19) (*Li et al., 2009*). A sliding window of 500 bp was used to calculate the average read depth over a given genome segment and normalized to the average read depth of the nuclear genome. Read depth analysis was visualized in R (v3.5.2) using ggplot2. Compiled read depth analysis found in *Supplementary file 6*.

## CNV breakpoint detection

Chromosomes containing CNVs were detected using the Yeast Mapping Analysis Pipeline (YMAP v1.0 [*Abbey et al., 2014*]). Fastq files were uploaded and mapped to the SC5314 reference genome (A21-s02-m08-r09) with correction for GC-content and chromosome end bias. Estimated copy number breakpoints were detected using Aneufinder (v1.10.2), with a bin width of 42.5 kb (*Bakker et al., 2016*). The bins containing estimated breakpoints were identified for further analysis. Estimated breakpoints located at the MRSs were not analyzed due to the repetitive nature and poor mapping of these genomic regions. To further refine copy number breaks, fastq files were aligned to the SC5314 reference genome (see above) and read depth was determined for each nucleotide in the genome (samtools depth `-aa`, v0.1.19). Read depth was normalized to the mean read depth of the nuclear genome using R (v3.5.2). The 42.5 kb windows containing estimated copy number breakpoints as determined by Aneufinder were further subdivided into 500 bp windows. The mean normalized read depth was determined for these 500 bp windows and a rolling mean of every two consecutive 500 bp windows was determined using R. Copy number breakpoints were identified if 75% of four consecutive 500 bp windows had a mean normalized read depth that deviated from the mean nuclear genome read depth by more than 25% (*Ford et al., 2015*). Boundaries were further confirmed by visual inspection in Integrative Genomics Viewer (IGV v2.3.92) (*Thorvaldsdóttir et al., 2013*). CNV breakpoint positions were compared to the list of long repeat sequences found across the *C. albicans* genome described in *Supplementary file 2* of *Todd et al., 2019* and breakpoints were assigned a repeat name if they fell within 2 kb of a long repeat sequence (*Todd et al., 2019*).

## Copy number detection of complex CNVs

For each isolate, read depth for each position within the genome was calculated using samtools depth (`-aa`) (v0.1.19) (*Li et al., 2009*). A sliding window of 500 bp was used to calculate the average read depth over a given genome segment and normalized to the average read depth of the nuclear genome. Each segment of the chromosome containing a complex CNV (high copy number central region and all lower copy number flanking sequences, including the disomic regions) was then assigned a copy number using the average normalized read depth of the 500 bp windows located within that given CNV segment multiplied by two for the diploid genome. If a 500 bp window contained a known repeat sequence (including those associated with copy number breakpoints) that window was excluded from analysis due to read mapping errors that occur in repetitive sequences. Likewise, telomere proximal regions were excluded from analysis. Telomere proximal regions were determined as the start or end of each chromosome to the first confirmed, non-repetitive genome feature as previously described (*Ene et al., 2018*; *Hirakawa et al., 2015*; *Todd et al., 2019*). The read depth for each segment was then normalized to the average read depth of the disomic segments of the chromosome to normalize for chromosome copy number, and rounded to the nearest integer for the final copy number. The copy number data, as well as read depth summary statistics for each chromosome segment are found in *Supplementary file 6*. Data were analyzed and summary statistics were generated using JMP Pro (v14.2.0).

## Allele ratio analysis

Heterozygous positions were determined using the Genomics Analysis Toolkit's HaplotypeCaller (v3.7–0-gcfed67) with a standard minimum confidence threshold of phred30 (-stand_call_conf 30) (*McKenna et al., 2010*). Variants were analyzed if the read depth was >2 and the variant was sequenced on both the forward and reverse strand. Additional filtering of SNVs included the removal of ancestral homozygous positions, homozygous SNVs that were maintained within the progenitor and all evolved isolates and not the reference SC5314. SNVs contained within long-repeat sequences were also removed due to the mapping errors with short read sequencing in repetitive regions.

## Variant calling

De novo variant detection was conducted using aligned, sorted BAM files that were converted to mpileup files using Samtools (samtools mpileup) (*Li et al., 2009*). VCFs were generated using Varscan (V2.3) (*Koboldt et al., 2012*). Called variants were filtered under the following conditions: 1) All SNVs shared by all progeny are removed and are assumed to be parentally derived, 2) a read depth of <5, and 3) a percent alternative allele of <0.2.

## Oxford Nanopore Technology MinION sequencing and de novo alignment

Two identical libraries of AMS3051 were constructed using the 1D Ligation Sequencing Kit (SQK-LSK108) from Oxford Nanopore following manufacturers protocol with slight modification. Briefly, end repair and dA-tailing was performed following New England Biolabs protocol for the Ultra II End-prep reaction (NEB E7546S) with a 30 min incubation at 20°C followed by a 30 min incubation at 65°C. The DNA was then purified using 1.8x Ampure beads (Agencourt). Adapter ligation was allowed to incubate at room temperature for 30 min followed by a 0.6x Ampure (Agencourt) bead cleanup. Data was generated using the R9 FLO-MIN106 spot-on flow cell. Calibration of the flow cell indicated 1171 active pores across the four mux groups. The library was loaded following manufacturers recommendation and sequencing was allowed to proceed for 24 hr before loading with the second library. Sequencing continued for another 24 hr before the sequencing run was terminated. A total of 910753 reads were obtained with a mean read length of 2210 bp (minimum 5 bp and maximum 352979 bp). Average theoretical coverage was 129.9x assuming a haploid genome size of 15.5 Mb.

## Visualization of Oxford Nanopore Technology MinION sequencing data

Oxford Nanopore minion Fastq files from AMS3051 were aligned to the SC5314 reference genome (A21-s02-m09-r08) using NGMLR (-x ont, v0.2.7) (*Sedlazeck et al., 2018*). The resulting SAM file was converted to a BAM file using samtools view (-S –b) and was sorted and indexed using samtools sort and samtools index, respectively (v0.1.19) (*Li et al., 2009*). Structural variant detection was conducted using Sniffles (v1.0.11) (*Sedlazeck et al., 2018*) and the average binned (10 kb) read coverage was determined using Copycat (https://github.com/MariaNattestad/copycat). Structural variants were identified using SplitThreader (http://splitthreader.com). Individual discordant reads were identified and visualized using Ribbon with a minimum alignment length of 1 kb (*Nattestad et al., 2016*).

## Acknowledgements

We thank Curtis Focht from our lab for assistance developing multiple bioinformatics pipelines, and Hung-ji Tsai, Laura Burrack, and Dana Davis for helpful discussions and feedback on the manuscript. We are grateful to Leah Cowen and her lab for the isolates detailed in *Figure 3*. Support for this research was provided by NIH grant R01 AI143689, NE Established Program to Stimulate Competitive Research (EPSCoR) First Award, NE Department of Health and Human Services (LB506-2017-55) award, and NIH-NCRR COBRE grant P20RR018788 sub-award. The sequencing datasets generated during this study are available in the Sequence Read Archive repository under BioProject PRJNA613282.

## Additional information

### Funding

| Funder | Grant reference number | Author |
|---|---|---|
| National Institute of Allergy and Infectious Diseases | AI143689 | Anna Selmecki |
| Nebraska Department of Health and Human Services | LB506-2017-55 | Anna Selmecki |
| Nebraska Established Program to Stimulate Competitive Research | First Award | Anna Selmecki |
| National Center for Research Resources | COBRE P20RR018788 sub-award | Anna Selmecki |

The funders had no role in study design, data collection and interpretation, or the decision to submit the work for publication.

### Author contributions

Robert T Todd, Conceptualization, Data curation, Formal analysis, Validation, Investigation, Visualization, Methodology, Writing - original draft, Writing - review and editing; Anna Selmecki, Conceptualization, Data curation, Formal analysis, Supervision, Funding acquisition, Visualization, Methodology, Writing - original draft, Writing - review and editing

### Author ORCIDs

Robert T Todd (iD) https://orcid.org/0000-0002-4522-7124
Anna Selmecki (iD) https://orcid.org/0000-0003-3298-2400

### Decision letter and Author response

Decision letter https://doi.org/10.7554/eLife.58349.sa1
Author response https://doi.org/10.7554/eLife.58349.sa2

## Additional files

### Supplementary files

- Supplementary file 1. Strains used in this study.
- Supplementary file 2. Features of long-repeat sequences identified at copy number breakpoints of complex CNVs.
- Supplementary file 3. Growth curve raw data and analysis.
- Supplementary file 4. Coding sequences within complex CNVs.
- Supplementary file 5. Primers used in this study.
- Supplementary file 6. Summary of complex CNV features.
- Transparent reporting form

### Data availability

All genome sequencing data have been deposited in the Sequence Read Archive under BioProject PRJNA613282 All data analyzed during this study are included in the manuscript and supporting files. The source data file is provided for Figure 2.

The following dataset was generated:

| Author(s) | Year | Dataset title | Dataset URL | Database and Identifier |
|---|---|---|---|---|
| Todd RT, Selmecki A | 2020 | Complex copy number variants in Candida albicans | https://www.ncbi.nlm.nih.gov/bioproject/? | NCBI BioProject, PRJNA613282 |

term=PRJNA613282

The following previously published dataset was used:

| Author(s) | Year | Dataset title | Dataset URL | Database and Identifier |
|---|---|---|---|---|
| Mount HO, Revie NM, Todd RT, Anstett K, Collins C, Costanzo M, Boone C, Robbins N, Selmecki A, Cowen LE | 2018 | Candida albicans genome sequencing | https://www.ncbi.nlm.nih.gov/bioproject/PRJNA323475/ | NCBI BioProject, PRJNA323475 |

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
