## [Decision Letter]

**Acceptance summary:**

We appreciate your efforts to address all the comments raised by the three reviewers, and we like how your work further unravels how medium-sized repeats in the *C. albicans* genome contribute to its capacity to quickly and reversibly adapt to certain stresses by stimulating amplification as well as loss of genomic regions.

**Decision letter after peer review:**

Thank you for submitting your article "Expandable and reversible copy number amplification drives rapid adaptation to antifungal drugs" for consideration by *eLife*. Your article has been reviewed by three peer reviewers, one of whom is a member of our Board of Reviewing Editors, and the evaluation has been overseen by Patricia Wittkopp as the Senior Editor. The following individual involved in review of your submission has agreed to reveal their identity: David Gresham (Reviewer #2).

The reviewers have discussed the reviews with one another and the Reviewing Editor has drafted this decision to help you prepare a revised submission.

We appreciate how this new study expands on your earlier work by showing how repeats are instrumental in facilitating transient amplifications of certain genomic loci that can help *C. albicans* become tolerant to drugs.

As you can see below, all three reviewers agree that your paper is interesting and well written, and we do not see a necessity for additional experiments. That said, all reviewers did mention that some points merit some additional clarification or discussion. Whereas some comments overlap, other points were only raised by one reviewer, likely reflecting the different expertise of the reviewers. However, we agree that all points that are raised are in fact valid and may help you to improve the paper and make it (even) more accessible and interesting to a broad readership. Hence, in this case, it seemed useful to copy all three separate reviews below instead of merging them all into one common file.

Reviewer #1:

In a previous study (*eLife* https://doi.org/10.7554/*eLife*.45954.001), this team showed how long repeat sequences scattered throughout the *C. albicans* genome fuel recombination events that lead to CNV and LOH. In this follow-up study, the authors show how subjecting *C. albicans* to antifungal drugs (fluconazole) leads to the selection of CNV in the form of large segmental duplications. The amplified regions ranged from 100 to 1000kb, often contain genes that have previously been associated with drug resistance, and some were amplified up to 12 times. Interestingly, the breakpoints always occurred within 2kb of long inverted repeat sequences, again suggesting that these repeats drive the recombination events. Moreover, removal of the antifungal drugs led to loss of the duplicated regions and return to the initial geno- and phenotype, suggesting that the extra copies impose a negative fitness effect the repeats also facilitate the loss of repeats.

Overall, whereas the idea of 1) CNVs as an adaptive process and 2) repeats as facilitators of genome instability are not new, this study is a useful follow up to the previously published work because it provides more mechanistic insight into the recombination events that allow adaptation to antifungals through swift amplification of certain chromosomal regions. Moreover, the study also shows that the amplification confers a fitness defect in environments without drugs, and how the amplification is quickly lost when the selective pressure disappears. Together, this reveals how medium-sized repeats in the *C. albicans* genome contribute to its capacity to quickly (and reversibly) adapt to certain stresses by stimulating amplification as well as loss of genomic regions.

Reviewer #2:

In this paper, Todd and Selmecki report copy number variants (CNVs) that arise in response to selection for antifungal resistance. The authors performed 48 evolution experiments in four different genetic backgrounds. Seven isolates that showed increased drug resistance were analyzed using DNA sequencing and CHEF gels. They identify high copy number amplifications of chromosomal segments in resistant clones that do not result from previously reported mechanisms (i.e. centromere amplifications, aneuploidy and use of major repeat sequences (MRS)). The novel CNVs have breakpoints that are near long repeat sequences and are characterized by an intrachromosomal symmetric stair-step pattern of copy number increase. The amplified regions contain several genes and the authors identify likely targets of selection including MRR1, which regulates drug efflux pump expression. Detailed analysis of one set of CNVs provides evidence for the formation of a dicentric region and repeat expansion through a breakage-fusion-bridge mechanism. They then show that when the selection is removed, CNVs rapidly revert.

This is an interesting study. Understanding mechanisms of antifungal resistance is of clear significance and this paper adds to our understanding of the diversity of mechanisms that can be used to generate CNVs underlying drug resistance in *C. albicans*. Although I do not think that the finding is sufficiently novel to warrant publication as a primary research paper in *eLife*, the report extends the initial publication by the authors on repeat sequences that mediate CNV formation in the *C. albicans* genome and therefore is consistent with the aims of the Research Advance format at *eLife*.

1) It is unclear to me the difference between a major repeat sequence (MJR) and a repeat sequence. The authors state that these breakpoints do not occur near MJR, but do occur "within 2 kb of a long repeat sequence". What is the difference?

2) The paper does not make clear the rate/frequency of this type of CNV forming mechanism. 48 evolution experiments were performed, but only seven clones were isolated for analysis. It's not clear if this report is part of a larger study and this is just a subset of the result. That should be made clearer and if it is part of a larger study the frequency of this mechanism, relative to other mechanisms that generated CNVs at these loci, reported.

3) The symmetrical stair-step property is really striking, but in some cases this seems imperfect (e.g. 2-4-6-5-3-2). How can the authors rule out that this isn't due to noise in the read depth analysis and this should actually be 2-4-6-4-2 or 2-3-5-3-2? i.e. what is the error on the copy number estimation? Is the proposed model expected to generate perfect symmetry?

4) It is surprising that the evolved strains do not have a growth rate defect in rich media. Does this suggest that there is not selection for the loss of the CNV, but simply that it occurs at a very high rate?

5) I do not understand the long read sequencing or its interpretation as it is poorly described. How many isolates were analyzed (the Materials and methods section indicates just one)? How can multiple structural variants be detected at a single locus (e.g. eight detected near repeat 127). I think the authors should improve the description and interpretation of these results.

6) The structure identified from long read sequencing sounds like that found for ODIRA breakpoints (reads switching from the complement to the reverse complement). In addition the stair-step structure is seen at some of the ODIRA-mediated CNVs in budding yeast. Can the authors exclude this DNA replication based mechanism as a possible mechanism forming these CNVs?

Reviewer #3:

In this Research Advance article, authors Todd and Selmecki identify and characterize several new types of fluconazole/azole-resistant isolates of *C. albicans* that evolve rapidly during growth in the presence of the drug(s) in vitro. Several of these isolates display a remarkable, novel mechanism of drug resistance and tolerance that involves amplification of the copy number of certain chromosome segments by a proposed breakage-fusion-bridge (BFB) model involving the repeated formation and repair of dicentric chromosome intermediates by non-allelic homologous recombination. This BFB model is consistent with the evidence presented and has been proposed to explain similar CNV amplification events in human cancers. The copy number amplification observed in this study is dependent on the presence of two sets of inverted long repeat sequences, which supports the Todd et al., 2019 publication in which previously unidentified long repeat sequences were shown to coincide frequently with breakpoints for CNV and LOH recombination events. This manuscript offers a robust, in-depth characterization of a new mechanism for rapid, adaptive (and reversible) genomic change that builds upon and expands our knowledge of the mechanisms contributing to drug resistance and genome plasticity in human fungal pathogens.

1) This study provides direct evidence of in vitro evolution by CNV amplification resulting in azole drug resistance/tolerance in *C. albicans*. One limitation of the study is that there is no evidence presented of *C. albicans* isolates evolving azole resistance via this mechanism in the context of a host infection. The authors could offer comments on how this model could be validated by either direct testing of clinical samples from patients treated with fluconazole or in a host model of infection. Also, in the context of a host infection, would there be any limitations to the formation of dicentric intermediates and breakage-fusion-bridge cycles (or would these events be more likely due to host-mediated DNA damaging agents)? Would cells survive and proliferate sufficiently to generate the copy number amplification necessary for resistance by this mechanism before being targeted for clearance by immune cells, particularly in the presence of fluconazole drug treatment that slows growth of the pathogen?

2) "Separation of Chrs 4-7 identified a dramatic increase in Chr4 size in two isolates (AMS4702 and AMS4444) with Chr4L CNVs relative to their progenitors and indicates that these CNVs are intra-chromosomal, rather than extra-chromosomal, amplifications (Figure 1—figure supplement 2)." In Figure 1—figure supplement 2, there is an absence of a band at Chr4 (denoted by the asterisk) compared to the progenitor for both isolates, but no visible "larger band" that would correspond to the noted "dramatic increase in Chr4 size for these isolates." Please explain (e.g. is the size of Chr4 too large to resolve/migrate through the gel?).

3) It is unclear in the text and figures (such as Figure 6) how the asymmetric stair-step copy numbers arise. A figure depicting symmetric vs. asymmetric stair-step copy number amplification would be helpful.

4) At least some work from budding and fission yeast should be cited (Symington, Kolodner, Lobachev), where genetic control has been examined in some cases. The model with inverted repeats usually involves some type of fold back invasion of an inverted repeat on the same chromosome, followed by BIR to the end, to generate a dicentric.

An example from Kolodner: Putnam et al., 2014.

Fold-backs have also been examined in fission yeast, and these initiated in the context of replication restart:

Mizuno et al., 2013

---

## [Author Response]

Reviewer #2:[…] This is an interesting study. Understanding mechanisms of antifungal resistance is of clear significance and this paper adds to our understanding of the diversity of mechanisms that can be used to generate CNVs underlying drug resistance in C. albicans. Although I do not think that the finding is sufficiently novel to warrant publication as a primary research paper in eLife, the report extends the initial publication by the authors on repeat sequences that mediate CNV formation in the C. albicans genome and therefore is consistent with the aims of the Research Advance format at eLife.1) It is unclear to me the difference between a major repeat sequence (MJR) and a repeat sequence. The authors state that these breakpoints do not occur near MJR, but do occur "within 2 kb of a long repeat sequence". What is the difference?

Thank you for indicating that this needed clarification. The Major Repeat Sequence (MRS) in *Candida albicans* is a well-characterized long track (10-100 kb) of nested DNA repeats, found 8 times throughout the genome on 7 of the 8 chromosomes. Previous work identified that the MRS is involved in karyotype rearrangements (Lephart and Magee, 2006). The repeats identified in Todd et al., 2019 are distinct from the MRS class of repeats, and were associated with CNVs, inversions, and loss of heterozygosity in a diverse collection of isolates. We have changed the following sentences to clarify the difference between the MRS and the long repeat sequences.

“All copy number breakpoints occurred within 2 kb of one of the 1974 long repeat sequences identified previously by Todd et al., 2019 (Supplementary file 2).”

“…and were not associated with any centromere (*CEN*) sequence or the *C. albicans* repetitive element known as the Major Repeat Sequence (MRS, found eight times within the *C. albicans* genome) (Chibana et al., 1994; Chindamporn et al., 1998; Lephart et al., 2006).”

2) The paper does not make clear the rate/frequency of this type of CNV forming mechanism. 48 evolution experiments were performed, but only seven clones were isolated for analysis. It's not clear if this report is part of a larger study and this is just a subset of the result. That should be made clearer and if it is part of a larger study the frequency of this mechanism, relative to other mechanisms that generated CNVs at these loci, reported.

Thank you for this suggestion. We are very interested in the rate/frequency of these events, especially given that genetically diverse clinical isolates can amplify recurrent CNVs (Figure 1) and that the breakpoints are hotspots for additional genome rearrangements like loss of heterozygosity and inversions in animal models (Todd et al., 2019). Because this is the first study to identify these complex CNVs in *C. albicans*, a robust, in-depth characterization the CNVs and the mechanisms driving their formation is a necessary first step. The seven isolates described are a part of a larger study into the mechanisms that drive antifungal drug resistance in *C. albicans,* however most of the remaining populations have not been analyzed yet by whole genome sequencing. Of the 48 FLC-evolved populations, only 14 have been sequenced and 7/14 have complex CNVs. We now include this frequency in the Results section (subsection “Extensive copy number amplifications occur during adaptation to antifungal stress”).

Future studies using single cell analyses with selectable markers are needed to determine the rate of CNV formation during antifungal drug selection and the frequency of these events relative to other adaptive mechanisms. We address the limitations of this study and the need for future experiments that directly quantify the rate of complex CNVs during antifungal drug selection in a new paragraph in the Discussion section (subsection “Model for Complex CNV formation”, second paragraph).

3) The symmetrical stair-step property is really striking, but in some cases this seems imperfect (e.g. 2-4-6-5-3-2). How can the authors rule out that this isn't due to noise in the read depth analysis and this should actually be 2-4-6-4-2 or 2-3-5-3-2? i.e. what is the error on the copy number estimation? Is the proposed model expected to generate perfect symmetry?

First, to clarify, we observed both symmetric (e.g. AMS4702) and asymmetric (e.g. AMS4105) stair-step amplifications. All symmetric CNVs were associated with two inverted repeat sequences, while the asymmetric CNVs were associated with three inverted repeat sequences. We propose that the asymmetric stair-step CNVs arise via a similar dicentric mechanism as symmetric CNVs, however instead of the dicentric chromosome breaking centromere-proximal it breaks within the chromosome arm and leads to NAHR between different repeat sequences. We added Figure 6—figure supplement 1 to describe an example of asymmetric stair-step formation.

Second, we agree that whole genome sequencing read depth data inherently includes some noise. To calculate the copy numbers, read depth for every position in the genome was determined using samtools. Average read depth was calculated using a 500 bp sliding window. The copy number was calculated by normalizing the average read depth to the average nuclear genome depth and multiplying by two. The copy number for each stair-step of the complex CNV was calculated by averaging the copy number for all 500 bp windows contained in each stair-step. To determine the error of the copy number of each stair-step, we calculated the standard error of the mean (SEM). The SEMs ranged from 0.005 – 0.383 (SEM was added to Supplementary file 6). To determine if the SEM was consistent across all stair-steps of the complex CNVs, we determined if there was a correlation between copy number and SEM or stair-step length and SEM. There was a positive correlation between copy number and SEM (p < 0.002, r_2_ = 0.21) and a negative correlation with stair-step length and SEM (p < 0.003, r_2_ = 0.19), indicating that short, high copy number stair-steps have a higher SEM than longer, lower copy number stair-steps (see example below). To determine if the calculated error would alter copy number, we determined the copy number of each stair-step +/- the SEM, and then normalized to the base diploid genome and then rounded to the nearest integer for biological relevance. Of the 46 copy number stairsteps (Supplementary file 6), only five stair-steps changed copy number upon the incorporation of the SEM. Three of these stair-steps were located between the same long inverted repeat sequence with a short spacer distance (~5 kb) on Chr3R. For example, AMS4104 Chr3R copy numbers +/- SEM were rounded to: 2-7-13-7-2 or 2-8-13-7-2. Importantly, the overall structure of the complex CNV was not altered, only the rounding of this single stair-step within the CNV. Therefore, we are confident in the copy numbers presented in the manuscript that form both symmetric and asymmetric CNVs.

4) It is surprising that the evolved strains do not have a growth rate defect in rich media. Does this suggest that there is not selection for the loss of the CNV, but simply that it occurs at a very high rate?

We agree that this is surprising, given the size (length and amplitude) of the complex CNVs. However, in general, whole chromosome aneuploidies and CNVs that comprise an entire arm of a chromosome (isochromosomes) are well-tolerated in *C. albicans* (Selmecki et al., 2006; Todd et al., 2019).

Accordingly, the growth rate data support that most of the isolates with a complex CNV do not have a growth rate defect in rich media. Importantly, these complex CNVs are presumed to be maintained on a monocentric chromosome (Figure 1). We do not know the rate in which these CNVs are lost, but there appears to be very little selection for loss based on growth rate. In contrast, isolates with a complex CNV on a dicentric chromosome (and a partial monosomy) have a strong growth defect (Figure 5). Fast growing clones are rapidly isolated from the dicentric isolate (after one plating on rich media) and these clones have lost the dicentric and the partial monosomic chromosomes (via recombination with the remaining full-length homolog). Interestingly, in the absence of drug selection, these fast growing clones include both euploid (WT) clones and clones containing complex CNVs maintained on a monocentric chromosome (like in Figure 1), which further supports that the cells with complex CNVs can compete with WT genotypes.

5) I do not understand the long read sequencing or its interpretation as it is poorly described. How many isolates were analyzed (the Materials and methods section indicates just one)? How can multiple structural variants be detected at a single locus (e.g. eight detected near repeat 127). I think the authors should improve the description and interpretation of these results.

Thank you for these suggestions, as mentioned above in response to reviewer 1: We have made significant changes to Figure 4 and the Figure 4 legend to clarify how the long-read sequencing was used to characterize novel recombination products (fold-back inversions and non-allelic homologous recombination between inverted repeats) in the azole evolved isolate, AMS3051. The only isolate sequenced using the Oxford Nanopore MinION was AMS3051 that contains the Chr3L complex CNV on a dicentric chromosome. We changed the color scheme and added a schematic to show the WT and post-recombination chromosomes. We also modified the Results section to better describe how the long-read sequences supported each of the two structural variants, one at Repeat 124 and one at Repeat 127 (subsection “Recombination occurs between long inverted repeats leading to CNV formation”).

6) The structure identified from long read sequencing sounds like that found for ODIRA breakpoints (reads switching from the complement to the reverse complement). In addition the stair-step structure is seen at some of the ODIRA-mediated CNVs in budding yeast. Can the authors exclude this DNA replication based mechanism as a possible mechanism forming these CNVs?

We agree that these two features (complement/reverse complement switching and stair-step patterns) are similar between the copy number breakpoints described in this manuscript and copy number breakpoints attributed to ODIRA. However, to the best of our knowledge, the other key features of ODIRA are not observed in our data. In the ODIRA model of CNV formation, errors during DNA replication result in the ligation of the leading and lagging strands at short (~8 bp), closely spaced (~40 bp) inverted repeat sequences within the same replication fork. In contrast, the long inverted repeat sequences associated with the complex CNVs in *C. albicans* are both much longer (median copy length of ~1.7 kb) and separated by a much greater distance (ranging from 1609 bp – 96,231 bp, median spacer distance ~38,000 bp) than the repeats observed in *S. cerevisiae*. Given the distance between the inverted repeat sequences in *C. albicans*, it is unlikely that the repeats would be present in the same replication fork and undergo ligation of the leading and lagging strands, as in ODIRA. In addition, we determined the distance between the *C. albicans* predicted origins of replication (Tsai et al., 2014) and the long repeat sequences associated with the complex CNVs and found that the nearest predicted origins were located between ~600 bp – ~100 kb (median distance ~16 kb) away from the long repeats. Therefore, while we cannot formally exclude a DNA replication based mechanism in the formation of *C. albicans* complex CNVs, the length and distance between inverted repeats, and distance between the repeats and origins of replication, makes this unlikely.

Reviewer #3:[…] 1) This study provides direct evidence of in vitro evolution by CNV amplification resulting in azole drug resistance/tolerance in C. albicans. One limitation of the study is that there is no evidence presented of C. albicans isolates evolving azole resistance via this mechanism in the context of a host infection. The authors could offer comments on how this model could be validated by either direct testing of clinical samples from patients treated with fluconazole or in a host model of infection. Also, in the context of a host infection, would there be any limitations to the formation of dicentric intermediates and breakage-fusion-bridge cycles (or would these events be more likely due to host-mediated DNA damaging agents)? Would cells survive and proliferate sufficiently to generate the copy number amplification necessary for resistance by this mechanism before being targeted for clearance by immune cells, particularly in the presence of fluconazole drug treatment that slows growth of the pathogen?

Thank you for suggesting additional discussion of our observations in the context of a host infection. We agree that a limitation of this study is that we do not know the rate of dicentric chromosome and complex CNV formation in vitro or in vivo. in vivo passage of *C. albicans* through a mouse model of candidiasis (in the absence of antifungal drug) also identified CNVs at long repeat sequences (Todd et al., 2019). This suggests that these long repeat sequences can generate genomic variation in diverse environments, including in vivo. Host factors, such as reactive oxygen species may increase the rate of DSBs that result in dicentric chromosome formation, in the presence or absence of antifungal drugs. Indeed, dicentric chromosomes have been identified in clinical isolates of *C. albicans*, suggesting that these chromosome structures can form in vivo (Selmecki, 2006). Future studies using single-cell analyses (e.g. Lauer et al., 2018) are needed to determine the rate of complex CNV formation both in vitro and in vivo in the presence and absence of antifungal drugs. We have added a new paragraph to the Discussion section to address these points (subsection “Model for Complex CNV formation”, second paragraph).

2) "Separation of Chrs 4-7 identified a dramatic increase in Chr4 size in two isolates (AMS4702 and AMS4444) with Chr4L CNVs relative to their progenitors and indicates that these CNVs are intra-chromosomal, rather than extra-chromosomal, amplifications (Figure 1—figure supplement 2)." In Figure 1—figure supplement 2, there is an absence of a band at Chr4 (denoted by the asterisk) compared to the progenitor for both isolates, but no visible "larger band" that would correspond to the noted "dramatic increase in Chr4 size for these isolates." Please explain (e.g. is the size of Chr4 too large to resolve/migrate through the gel?).

Thank you for suggesting additional details for this figure. The wild-type Chr4 in *C. albicans* is ~1.6 MB in length. Given the size of complex CNVs in AMS4702 and AMS4444, the estimated size of Chr4 increases to ~3.4 MB and ~2.8 MB. These larger bands co-migrate with Chr1 and ChrR. To address this, we added the chromosome sizes to the CHEF gel image in Figure 1—figure supplement 2, and updated the figure legend.

3) It is unclear in the text and figures (such as Figure 6) how the asymmetric stair-step copy numbers arise. A figure depicting symmetric vs. asymmetric stair-step copy number amplification would be helpful.

Thank you for this suggestion. Asymmetric stair-steps were associated with three inverted repeat sequences, whereas symmetric stair-steps were associated with only two inverted repeat sequences. We propose that the asymmetric stair-step CNVs arise via a similar dicentric mechanism as symmetric CNVs, however instead of the dicentric chromosome breaking centromere-proximal it breaks within the chromosome arm and leads to NAHR between different repeat sequences. We have added Figure 6—figure supplement 1 as an example of asymmetric stair-step formation.

4) At least some work from budding and fission yeast should be cited (Symington, Kolodner, Lobachev), where genetic control has been examined in some cases. The model with inverted repeats usually involves some type of fold back invasion of an inverted repeat on the same chromosome, followed by BIR to the end, to generate a dicentric.An example from Kolodner: Putnam et al., 2014.Fold-backs have also been examined in fission yeast, and these initiated in the context of replication restart: Mizuno et al., 2013.

Thank you for these suggestions, we agree that fold back invasion of a repeat sequence on the same chromosome followed by BIR is as likely as NAHR between repeat sequences on sister chromatids. Because we cannot distinguish between these two events, we now indicate that both are possible in the Discussion (subsection “Model for Complex CNV formation”, first paragraph) and in the Figure 6 legend, and include the above references accordingly. One interesting distinction is that the long inverted repeats described in this study are separated by greater distances (median spacer distance ~38 kb) than repeat sequences typically involved in fold back invasion and BIR in *S. cerevisiae* and *S. pombe*.